# PKC-phosphorylation of Liprin-α3 triggers phase separation and controls presynaptic active zone structure

Javier Emperador-Melero[1], Man Yan Wong[1], Shan Shan H. Wang[1], Giovanni de Nola[1], Hajnalka Nyitrai[1,3], Tom Kirchhausen[2] & Pascal S. Kaeser [1✉]

The active zone of a presynaptic nerve terminal defines sites for neurotransmitter release. Its protein machinery may be organized through liquid–liquid phase separation, a mechanism for the formation of membrane-less subcellular compartments. Here, we show that the active zone protein Liprin-α3 rapidly and reversibly undergoes phase separation in transfected HEK293T cells. Condensate formation is triggered by Liprin-α3 PKC-phosphorylation at serine-760, and RIM and Munc13 are co-recruited into membrane-attached condensates. Phospho-specific antibodies establish phosphorylation of Liprin-α3 serine-760 in transfected cells and mouse brain tissue. In primary hippocampal neurons of newly generated Liprin-α2/α3 double knockout mice, synaptic levels of RIM and Munc13 are reduced and the pool of releasable vesicles is decreased. Re-expression of Liprin-α3 restored these presynaptic defects, while mutating the Liprin-α3 phosphorylation site to abolish phase condensation prevented this rescue. Finally, PKC activation in these neurons acutely increased RIM, Munc13 and neurotransmitter release, which depended on the presence of phosphorylatable Liprin-α3. Our findings indicate that PKC-mediated phosphorylation of Liprin-α3 triggers its phase separation and modulates active zone structure and function.

[1] Department of Neurobiology, Harvard Medical School, Boston, MA, USA. [2] Departments of Cell Biology and Pediatrics, Harvard Medical School and Program in Cellular and Molecular Medicine, Boston Children's Hospital, Boston, MA, USA. [3] Present address: VIB-KU Leuven Center for Brain and Disease Research, Campus Gasthuisberg, Leuven, Belgium. ✉email: kaeser@hms.harvard.edu

Membrane-free subcellular compartments can form through liquid–liquid phase separation, a process in which multivalent, low affinity interactions enable de-mixing of proteins into liquid condensates[1–3]. These condensates maintain high local protein concentrations and enable dynamic rearrangements and exchange of protein with the environment. Past work has established that protein complexes for many cellular processes, ranging from gene transcription to synaptic transmission, can be organized as phase condensates. It has remained challenging, however, to establish how phase separation controls intracellular functions.

Within a synapse, neurotransmitter release is restricted to specialized presynaptic structures called active zones[4,5]. These membrane-attached, dense scaffolds are formed by the multidomain proteins RIM, Munc13, RIM-BP, Piccolo/Bassoon, ELKS, and Liprin-α, and are essential for the sub-millisecond precision of synaptic vesicle exocytosis. While many release mechanisms are established, key questions on how these dense scaffolds assemble, and how they remain dynamic to support the high spatiotemporal demands of the synaptic vesicle cycle remain unanswered. Purified RIM1 and RIM-BP2 form liquid condensates in vitro, indicating that active zones may assemble following phase transition principles[6,7]. Other synaptic compartments may also be organized through phase separation[8–11]. Whether phase separation occurs at synapses in vivo and whether it is important for controlling synaptic release, however, remains uncertain.

Liprin-α proteins have received particular attention as assembly molecules because they control presynaptic structure of invertebrate synapses[12–15]. They contain N-terminal coiled-coils with Liprin-α homology (LH) regions and three C-terminal sterile alpha motifs (SAM domains)[5,13,16,17]. Mammals have four genes (Ppfia1-Ppfia4) that encode Liprin-α1 to Liprin-α4[16], of which Liprin-α2 and Liprin-α3 are strongly expressed in the brain and co-localize with active zone markers[18,19]. shRNA knockdown of Liprin-α2[20] or genetic deletion of Liprin-α3[19] causes loss of presynaptic proteins, similar to assembly defects after ablation of the single invertebrate gene[12–15,21]. While these data implicate Liprin-α in active zone assembly, the vertebrate Liprin-α functions, and their underlying mechanisms are not clear. Liprin-α2 and Liprin-α3 are not lost from synapses after genetic disruption of vertebrate active zones[19,22], which may reflect an upstream assembly function similar to invertebrates[14,15], or suggest that Liprin-α proteins are not part of the same protein complex. An upstream function aligns well with the broad interaction repertoire of Liprin-α, which includes active zone proteins, motors, cell adhesion proteins, and cytoskeletal elements[5,15,16,23–27]. Liprin-α interactions are further regulated by phosphorylation[28], making these proteins candidate effectors of kinase pathways that control exocytosis, for example of protein kinase A (PKA), phospholipase C (PLC)/protein kinase C (PKC), or Ca$^{2+}$/calmodulin-dependent kinase II (CaMKII) signaling[29]. In aggregate, previous data suggest that Liprin-α may connect active zone assembly to pathways for synapse development and plasticity.

Here, we find that PKC phosphorylation of serine-760 (S760) of Liprin-α3 rapidly triggers Liprin-α3 phase separation. RIM and Munc13-1, two important active zone proteins, are co-recruited into plasma membrane attached phase condensates, reminiscent of active zone formation. Ablation of Liprin-α2 and Liprin-α3 using double knockout mice that were generated for this study leads to reduced levels of RIM and Munc13-1 at synapses, impaired vesicle docking and a decreased pool of readily releasable vesicles. Abolishing Liprin-α3 phosphorylation via a single point mutation prevents its phase separation and its ability to reverse defects in active zone structure and in the pool of releasable vesicles. Similarly, we discover a rapid increase of RIM and Munc13-1 at the active zone upon activation of PKC, which

necessitates Liprin-α3 phosphorylation. We propose a working model in which active zone structure is dynamically modulated by Liprin-α3 phase condensation under the control of PKC.

## Results

### Liprin-α3 rapidly undergoes phase separation under the control of PLC/PKC signaling in transfected HEK293T cells.
Because Liprin-α3 is regulated by phosphorylation and controls active zone assembly[13,19,28], we asked whether Liprin-α3 is modulated by kinase pathways to control release site structure. Prominent presynaptic pathways operate via PKA, PLC/PKC, and CaMKII signaling[29]. We expressed mVenus-tagged Liprin-α3 in HEK293T cells and investigated whether activation or inhibition of these pathways alters Liprin-α3 distribution. Under basal conditions, mVenus-Liprin-α3 is predominantly soluble. Strikingly, after addition of the diacylglycerol analog phorbol 12-myristate 13-acetate (PMA), Liprin-α3 formed spherical condensates within minutes (Fig. 1a, Supplementary Fig. 1a, and Supplementary Movie 1). PMA mimics PLC-induced generation of diacylglycerol and activates PKC, suggesting that Liprin-α3 may be phosphorylated by PKC. This effect was not observed for other manipulations, including inhibition of PKC, or activation or inhibition of PKA or CaMKII (Fig. 1a and Supplementary Fig. 1b). The reorganization of Liprin-α3 into droplets occurred in all cells within minutes, was reversible upon washout, and droplet formation was independent of the mVenus-tag (Fig. 1b, c and Supplementary Fig. 1c, d).

Formation of spherical droplets is indicative of liquid–liquid phase separation[1,2]. Principles of liquid dynamics and phase separation predict that liquid condensates dynamically exchange molecules with the cytosol, undergo fusion and fission, relax into spherical shapes, and lack membrane boundaries between the condensed phase and the cytosol[1,2]. First, to test exchange of molecules, we assessed fluorescence recovery after photobleaching (FRAP), as implemented before to study liquid phases of synaptic proteins[6–11]. Individual condensates recovered to ~40% of the initial fluorescence at a fast rate ($t_{1/2 \, max \, recovery} = 16.1 \pm 0.4$ s), and a second bleaching of the same condensates resulted in near-complete recovery at a similar rate ($t_{1/2 \, max \, recovery} = 15.8 \pm 0.5$ s), establishing that the mobile fraction remains fully mobile (Fig. 1d, e). Second, we detected fusion and fission reactions of Liprin-α3 droplets (Supplementary Fig. 1e, f and Supplementary Movie 1). Droplet fusion was followed by exponential relaxation (Fig. 1f, g), as expected for liquid condensates[30,31]. Third, to assess whether these fluorescent droplets are indeed membrane-free condensates, we used correlative light-electron microscopy (CLEM). Liprin-α3 condensates formed electron dense structures that were not surrounded by lipid bilayers (Fig. 1h, i). We conclude that Liprin-α3 forms liquid condensates as a function of PLC/PKC signaling.

### PKC phosphorylates Liprin-α3 at S760.
We hypothesized that PMA triggers PKC activation followed by phosphorylation of Liprin-α3 to induce phase separation. To investigate whether Liprin-α3 is a PKC substrate, we purified GST-fusion proteins covering the entire Liprin-α3 protein, and incubated them with $^{32}$P-labeled ATP and recombinant PKC (Fig. 2a, b). The linker region between the N-terminal Liprin homology regions and the C-terminal SAM domains most efficiently incorporated $^{32}$P, and mass spectrometry identified five phosphorylated serine residues (S650, S751, S760, S763, and S764, Supplementary Fig. 2a). Notably, S760, but not other residues, was surrounded by a PKC consensus sequence[32], and this sequence was conserved in rat, human, and mouse Liprin-α3 (Supplementary Fig. 2b). In other Liprin-α proteins, a glycine residue was present instead of serine

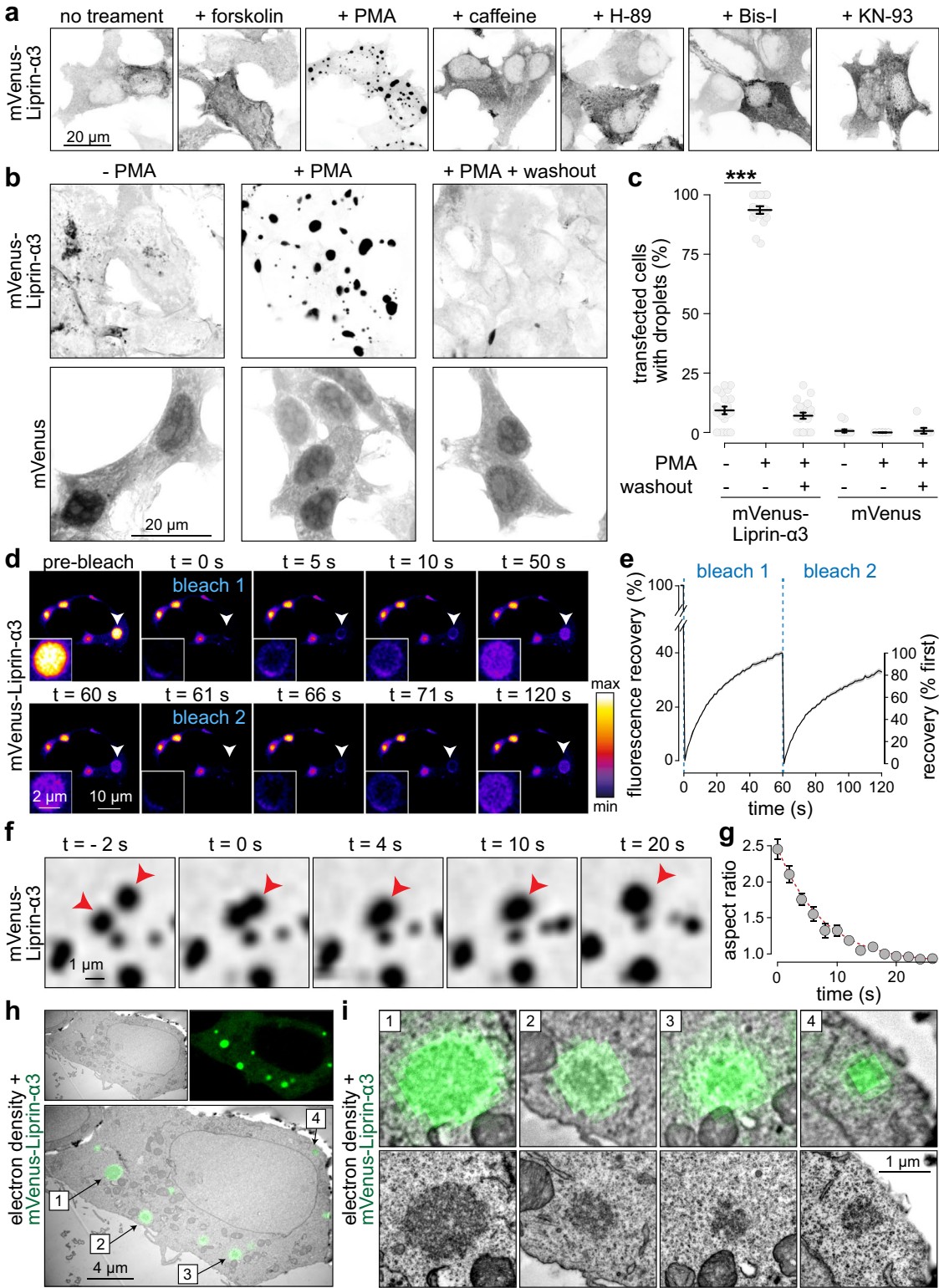

at the corresponding position. To determine whether any of these residues is responsible for phase transition, we engineered point mutations in mVenus-Liprin-α3 to abolish phosphorylation and expressed these constructs in HEK293T cells. S760A and S764A Liprin-α3 were incapable of PMA-induced droplet formation, while other point mutations did not impair it (Supplementary Fig. 2c).

To test whether these residues are phosphorylated, we generated anti-phospho-S760 and -S764 Liprin-α3 antibodies. The antibodies

detected a band at ~150 kDa in transfected HEK293T cells. Upon PMA addition, anti-phospho-S760-Liprin-α3 antibody signals increased, and disappeared when co-incubated with PKC blockers (Fig. 2c). Anti-phospho-S764 antibody signals were unaffected by the same manipulations (Supplementary Fig. 2d). Furthermore, when we incubated a recombinant Liprin-α3 fragment with PKC, phospho-S760 antibodies detected a signal increase while phospho-S764 did not (Supplementary Fig. 2e), making it unlikely that S764 is directly phosphorylated by PKC. Phospho-S760 Liprin-α3 was

**Fig. 1 Liprin-α3 undergoes liquid–liquid phase separation upon activation of PLC/PKC signaling pathways. a** Confocal images of fixed HEK293T cells transfected with mVenus-Liprin-α3, without treatment or in the presence of forskolin (to activate PKA), PMA (to activate PLC/PKC), caffeine (to activate CamKII), H-89 (to inhibit PKA), bisindolylmaleimide-I (Bis-I, to inhibit PKC), or KN-93 (to inhibit CamKII), representative cells from two to three overview images (containing several cells for each condition, one transfection each) are shown. **b, c** Example confocal images (**b**) and quantification of % cells containing droplets (**c**) of fixed HEK293T cells transfected with mVenus-Liprin-α3 or mVenus alone. Cells were fixed 15 min after PMA addition or 6 h after washout, N = 21 images/3 independent batches of cells each for mVenus-Liprin-α3, 18/3 each for mVenus, p values: mVenus-Liprin-α3 (tested against −PMA), +PMA 8e-8 (***), washout 0.58; mVenus, 0.82. **d, e** Example time-lapse images (**d**) and quantification (**e**) of the fluorescence recovery after photobleaching (FRAP) of mVenus-Liprin-α3 condensates in live, transfected HEK293T cells. Two consecutive bleach steps were applied, N = 30 droplets/ 3 independent transfections. **f, g** Example live time-lapse confocal images of a Liprin-α3 condensate undergoing fusion (**f**) and quantification of the aspect ratio over time (**g**) as a measure of relaxation after fusion, N = 10 fusion events, the exponential fit (red dotted line) is described with the formula $AR = 0.93 + 1.48 * \exp(-t/7.41)$. **h, i** Correlative light-electron microscopy (CLEM) example image of a fixed HEK293T cell transfected with mVenus-Liprin-α3 and incubated with PMA showing an overview with multiple condensates (**f**) and individual droplets (**g**) magnified from the overview image (top) or independently acquired higher magnification images of the same droplets (bottom), a representative cell from a total of three cells (one transfection) that were assessed by CLEM is shown. Summary data in **c, e**, and **g** are mean ± SEM. Significance was assessed using Kruskal–Wallis tests with Holm post hoc comparison between all groups, and only results compared to the respective −PMA condition are reported in **c**. All tests were two-sided. For a time course of phase separation, phase separation of non-tagged Liprin-α3, and fusion and fission of Liprin-α3 condensates, see Supplementary Fig. 1 and Supplementary Movie 1.

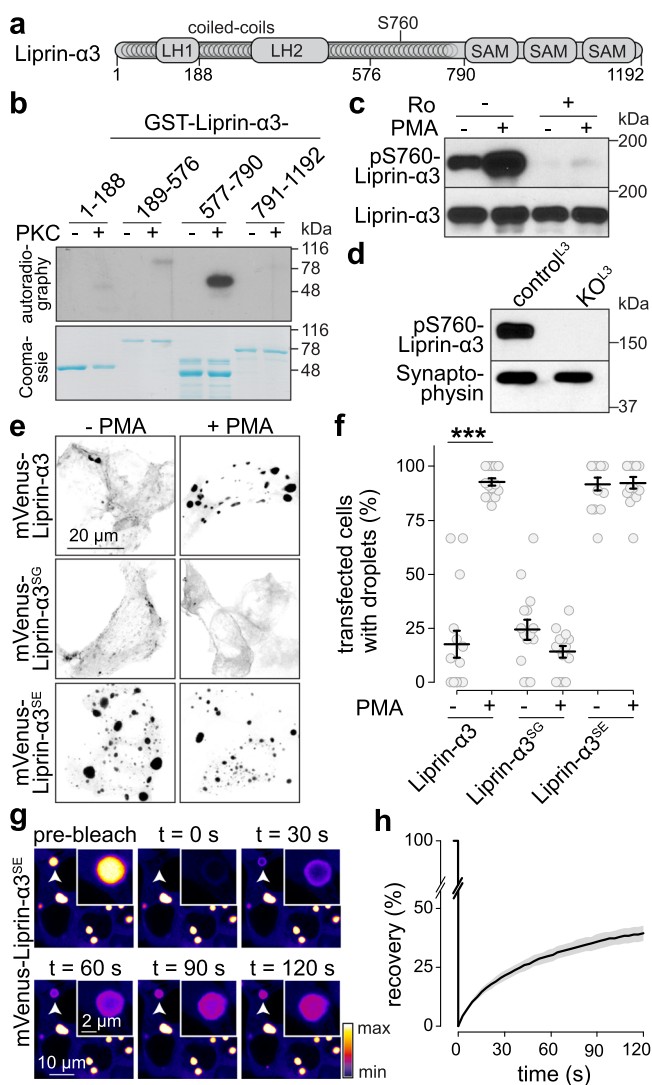

**Fig. 2 Protein kinase C phosphorylation of Liprin-α3 at S760 induces phase separation. a** Schematic of the rat Liprin-α3 domain structure showing Liprin homology regions 1 and 2 (LH-1 and -2), coiled-coil regions, and sterile alpha motifs (SAM). **b** Autoradiography (top) and Coomassie staining (bottom) of purified GST-Liprin-α3 fragments incubated with $^{32}$P-γ-ATP and with or without recombinant PKC, representative scans from two pairs of blots/Coomassie gels are shown. **c** Western blot of lysates of transfected HEK293T cells expressing mVenus-Liprin-α3, incubated with PMA and/or the PKC inhibitor Ro31-8220 (Ro, 1 μm), and immunoblotted with anti-phospho-S760 Liprin-α3 or Liprin-α3 antibodies that were generated for this study, scans from a single qualitative experiment are shown. **d** Western blot of lysates of cultured hippocampal neurons from Liprin-α3 knockout mice (KO$^{L3}$) or from heterozygote control mice (control$^{L3}$), scans from a single qualitative experiment are shown.

**e, f** Example confocal images (**e**) and quantification (**f**) of droplet formation in fixed HEK293T cells expressing mVenus-tagged Liprin-α3, phospho-dead S760G (Liprin-α3$^{SG}$), or phospho-mimetic S760E (Liprin-α3$^{SE}$) Liprin-α3, N = 15 images/3 independent transfections for Liprin-α3 and -α3$^{SG}$, N = 14/ 3 for Liprin-α3$^{SE}$, p values: Liprin-α3, 0.00004 (***); Liprin-α3$^{SG}$, 0.39; Liprin-α3$^{SE}$, 1.00. **g, h** Example live, time-lapse images (**g**) and quantification (**h**) of FRAP of mVenus-Liprin-α3$^{SE}$ condensates in transfected HEK293T cells. N = 14 droplets/3 independent transfections. Data were mean ± SEM. Significance was assessed using Kruskal–Wallis tests with Holm post hoc comparisons between all groups, and only results compared to the respective −PMA condition are reported in **f**. All tests were two-sided. For evaluation of additional potential phosphorylation sites, expression profile of phospho-S760 Liprin-α3 across brain areas and development, see Supplementary Fig. 2. For phase separation properties and IDR analyses of Liprin-α proteins, see Supplementary Fig. 3. For phase separation when Liprin-α2 and Liprin-α3 are co-expressed, see Supplementary Fig. 4. Source data for b–d are provided in the Source Data file.

most common residue at the +4 position at PKC target sites is a serine, and this may be part of the consensus[32]. Alternatively, S764 may participate in molecular rearrangements that lead to phase separation of Liprin-α3. In vivo, phospho-S760 Liprin-α3 signals were detected in the frontal cortex, hippocampus, cerebellum, and brain stem with high perinatal levels that gradually decreased over time (Supplementary Fig. 2f). We conclude that PKC phosphorylates S760 of Liprin-α3 in vitro and in vivo.

**S760 phosphorylation triggers liquid–liquid phase separation of Liprin-α3 in transfected HEK293T cells.** We established that

not detected in Liprin-α3 knockout neuronal cultures (Fig. 2d), confirming antibody specificity. Together, these experiments establish that PKC phosphorylates Liprin-α3 at S760, but likely not at S764. It is possible that the S764 mutation abolishes S760 phosphorylation (and Liprin-α3 phase separation) indirectly, as the

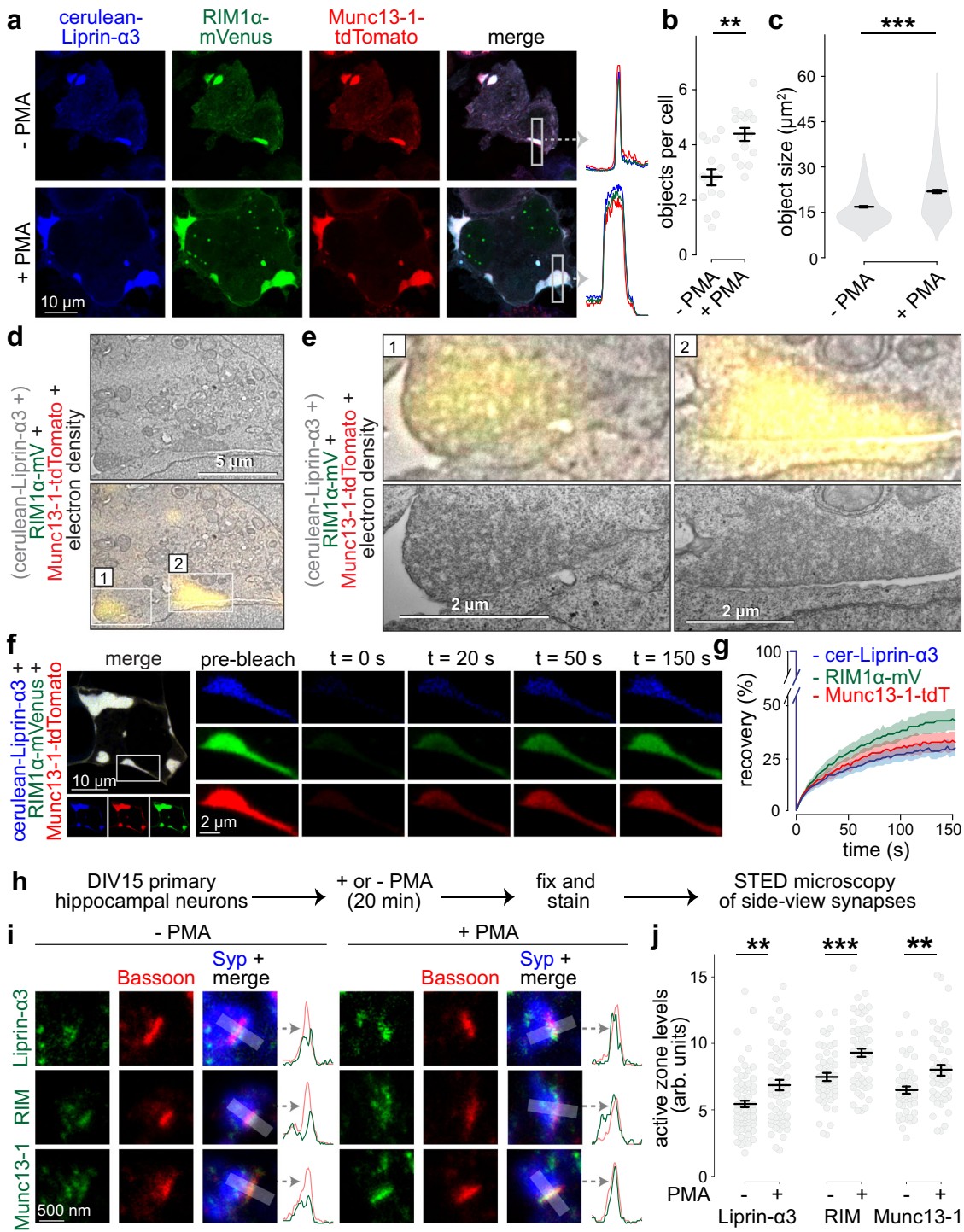

PKC phosphorylates Liprin-α3 at S760. To test whether this site mediates phase separation, we generated phospho-dead (S760G Liprin-α3$^{SG}$, using S→G substitution to make it similar to other Liprin-α proteins, Supplementary Fig. 2b) and phospho-mimetic (S760E, Liprin-α3$^{SE}$) mutants. Liprin-α3$^{SG}$ abolished PKC-induced phase separation, and Liprin-α3$^{SE}$ formed constitutive condensates independent of PKC activation (Fig. 2e, f) that showed dynamic exchange of molecules with the cytosol (Fig. 2g, h, $t_{1/2\ recovery} = 27.8 \pm 0.9$ s). We conclude that phosphorylation of S760 triggers liquid–liquid phase separation of Liprin-α3.

To evaluate whether phase separation may be shared by other vertebrate Liprin-α proteins, we first assessed whether intrinsically disordered regions (IDRs), which often drive phase condensation[2,33],

are detected. Mouse, rat, and human Liprin-α3 contained sequences reminiscent of IDRs at residues ~580 to ~790 (Supplementary Fig. 3a, b). Similarly, Liprin-α1, -α2, and -α4 also contained IDR sequences in the same region (Supplementary Fig. 3c). To test whether Liprin-α1, -α2, and -α4 undergo phase separation, we generated fluorophore-tagged versions of each of these proteins and assessed their distribution in transfected HEK293T cells. Liprin-α1 and -α4 appeared predominantly soluble, while Liprin-α2 had a droplet-like pattern (Supplementary Fig. 3d, e). Mimicking PLC/PKC signaling did not affect the distributions of Liprin-α1, -α2, or -α4, in line with a Liprin-α3-specific presence of S760. Furthermore, Liprin-α2 condensates did not recover after photobleaching (Supplementary Fig. 3f, g), indicating that they do not adopt liquid

**Fig. 3 Liprin-α3, Munc13, and RIM are co-recruited into phase condensates at the plasma membrane. a–c** Example confocal images including line profiles of highlighted regions (**a**) and quantification (**b**) of phase condensates in fixed HEK293T cells transfected with cerulean-Liprin-α3, RIM1α-mVenus, and Munc13-1-tdTomato in the absence or presence of PMA. Quantification of the number (**b**) and size (**c**) of protein condensates is shown, $N = 276$ (−PMA) or 405 (+ PMA) objects/15 images/3 independent transfections, p values: **b**, 0.0011 (**); **c**, 2e-16 (***). **d, e** CLEM example images of a fixed HEK293T cell transfected with cerulean-Liprin-α3, RIM1α-mVenus, and Munc13-1-tdTomato and incubated with PMA showing an overview (**d**) and detailed individual condensates (**e**) magnified from the overview image (**e**, top), and independently acquired images at higher magnification of the same condensates (**e**, bottom). Cerulean-Liprin-α3, which is consistently recruited to RIM/Munc13-containing condensates (**a**), is present in the transfection but not displayed because the fluorescence microscope for CLEM lacked a laser for cerulean excitation, a representative cell from two cells (two transfections) that were assessed by CLEM is shown. **f, g** Example of FRAP experiment (**f**) and quantification (**g**) of droplets in live HEK293T cells transfected with cerulean-Liprin-α3 (cer-Liprin-α3), RIM1α-mVenus and Munc13-1-tdTomato, $N = 19$ droplets/3 independent transfections. **h** Schematic of the assessment of effects of PKC activation on active zone assembly. **i, j** Example STED images (**i**) and quantification (**j**) of the intensity of endogenous Liprin-α3, RIM, and Munc13-1 at the active zone. Synapses in side-view were identified by the active zone marker Bassoon (imaged by STED microscopy) aligned at the edge of a synaptic vesicle cluster marked by Synaptophysin (Syp, imaged by confocal microscopy). Corresponding STED intensity profiles are shown on the right of each image set. Peak intensities were measured in these intensity profiles and plotted in **j**, shown as arbitrary units (arb. units). Liprin-α3: $N = 71$ synapses/ 3 independent cultures (−PMA) and 65/3 (+PMA); RIM: $N = 55/3$ (−PMA) and 54/3 (+PMA); Munc13-1 $N = 46/3$ (−PMA) and 47/3 (+PMA), p values: Liprin-α3, 0.0027 (**); RIM, 0.00017 (***); Munc13-1, 0.0022 (**). Data were shown as mean ± SEM. Significance was assessed using two-sided Mann–Whitney rank sum tests in **b**, **c**, and **j**. For assessment of single and double transfections, condensate formation in the presence of PKC inhibitors, and FRAP without PMA treatment see Supplementary Fig. 5. For STED analysis workflow, peak positions of each protein, and assessment of Liprin-α3 levels using an independent antibody, see Supplementary Fig. 6.

properties. Hence, only Liprin-α3 is phosphorylated by PKC to undergo liquid–liquid phase separation. The finding that IDRs are present in all Liprin-α proteins between the LH and SAM regions suggest that the IDR alone is not sufficient to drive phase separation. Consistent with this prediction, none of the individual Liprin-α3 fragments, including a fragment encompassing residues 577–790 that contains the IDR, underwent phase condensation with or without PMA (Supplementary Fig. 3h, i).

We finally assessed whether co-expression of the synaptic Liprin-α proteins, Liprin-α2, and Liprin-α3[19,34] influenced the condensate-behavior of these proteins in HEK293T cells. Liprin-α3 was recruited to the periphery of the Liprin-α2 condensate-like structures, the addition of PMA increased the formation of condensates containing Liprin-α2 and -α3, and Liprin-α3 but not -α2 showed dynamic exchange of proteins with the surrounding cytosol (Supplementary Fig. 4).

In summary, these in vitro data suggest that inter- or intramolecular interactions between several Liprin-α3 areas and potentially with other Liprin-α proteins may drive its phase separation, consistent with the model that such interactions drive Liprin-α function[35].

**Liprin-α3, RIM1α, and Munc13-1 are co-recruited into membrane-attached liquid condensates in transfected HEK293T cells.** If phase separation of Liprin-α3 participates in the control of active zone structure, active zone proteins should interact with Liprin-α3 liquid phases. Co-expression of cerulean-Liprin-α3 with either RIM1α-mVenus or Munc13-1-tdTomato in HEK293T cells resulted in recruitment of each protein into PMA-induced condensates (Supplementary Fig. 5a). Discrete, PMA-insensitive condensates were also observed when RIM1α was expressed alone (Supplementary Fig. 5b), in agreement with its intrinsic ability to phase separate[6]. Munc13-1 did not form droplets on its own, but PMA-dependent membrane recruitment was observed as previously described[36–38].

Co-expression of cerulean-Liprin-α3 with both RIM1α-mVenus and Munc13-1-tdTomato in HEK293T cells resulted in large protein condensates, and the addition of PMA increased their number and size (Fig. 3a–c). Remarkably, these condensates were not distributed throughout the cytosol, different from Liprin-α3 phase condensates, but were instead in close proximity to the plasma membrane. They persisted in the presence of PKC inhibitors, unlike condensates formed by Liprin-α3 alone (Supplementary Fig. 5c, d), reflecting their

constitutive formation. To assess whether they were membrane-attached, we used CLEM on PMA-treated cells. The fluorescent signals were highly overlapping with large protein densities that were not enclosed by membranes, but were adjacent to the plasma membrane, and their electron density merged with that of the lipid bilayer on the membrane-proximal side of the condensate (Fig. 3d, e).

We finally used FRAP to assess turnover of Liprin-α3, RIM1α, and Munc13-1 in these condensates. All three proteins rapidly recovered when the entire condensate was bleached (Fig. 3f, g) or when only small areas within large condensates were bleached (Supplementary Fig. 5e). Hence, condensates containing Liprin-α3, RIM1α, and Munc13-1 follow liquid dynamics. Overall, these data establish that Liprin-α3, RIM1α, and Munc13-1 coexist in protein-dense liquid condensates that appear attached to the plasma membrane, and formation of these condensates is enhanced by PLC/PKC signaling.

**PLC/PKC signaling increases active zone levels of Liprin-α3, RIM, and Munc13-1 at synapses of primary mouse hippocampal neurons.** Our findings suggest that activating PKC induces the formation of active zone-like, membrane-bound liquid condensates in transfected cells. If physiologically relevant, activation of this pathway should result in changes in active zone protein complexes at synapses. To test this, we assessed active zone levels of endogenous Liprin-α3, RIM, and Munc13-1 at synapses of cultured hippocampal neurons using stimulated emission depletion (STED) microscopy (Fig. 3h–j). As described previously[19,39–41], we analyzed synapses in side-view. These synapses were identified by the position of a bar-shaped active zone (marked by Bassoon, imaged in STED mode) relative to a synaptic vesicle cloud (identified by Synaptophysin, imaged in confocal mode), and the peak levels of proteins of interest were assessed within 100 nm of the Bassoon peak (see Supplementary Fig. 6a for an outline of synapse selection and analyses). Liprin-α3, RIM and Munc13-1 were predominantly clustered at the active zone with peak intensities falling within 50 nm from the peak of Bassoon (Supplementary Fig. 6b). Addition of PMA produced a 20–30% increase in peak active zone levels of Liprin-α3 (tested with two independent antibodies, Fig. 3i, j and Supplementary Fig. 6f–h), RIM and Munc13-1 (Fig. 3i, j) without affecting Bassoon (Supplementary Fig. 6c–e). Hence, mimicking PLC/PKC activation enhances active zone recruitment of RIM, Munc13-1, and Liprin-α3.

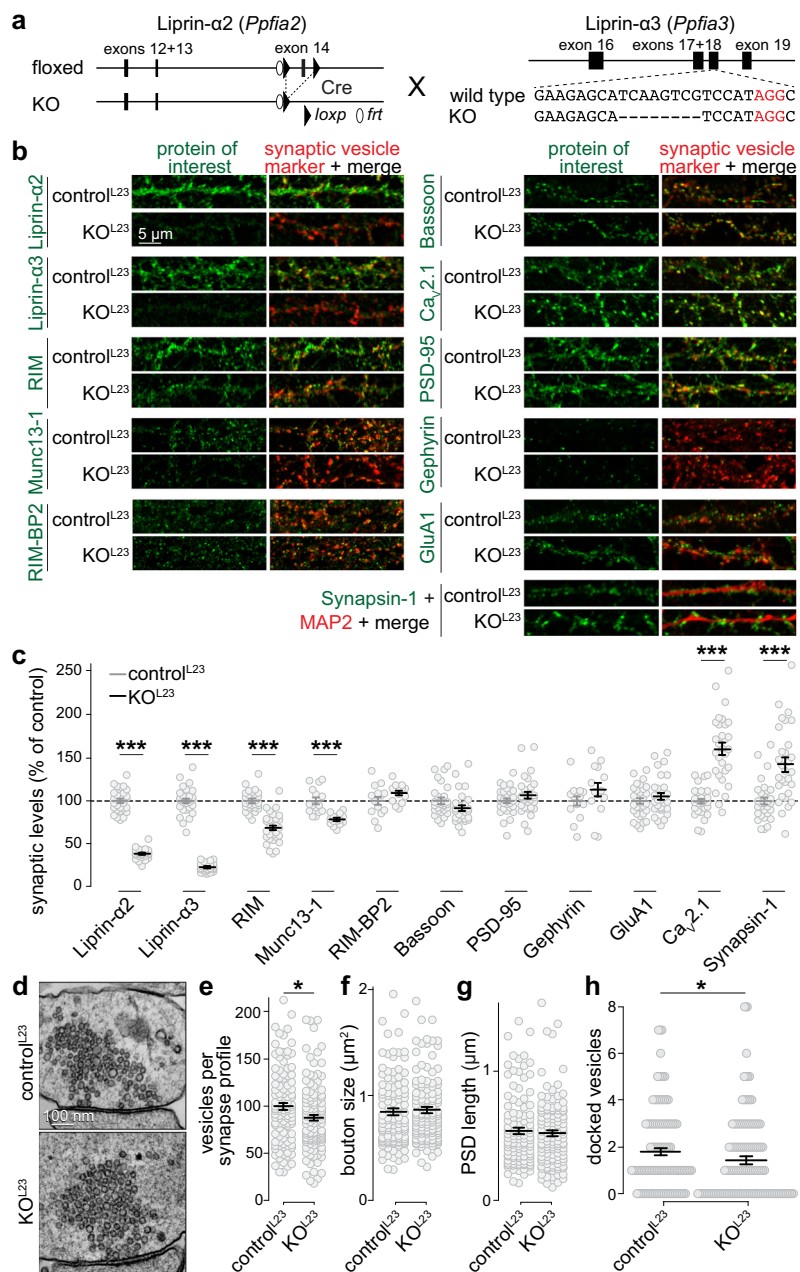

**Fig. 4 Liprin-α2/α3 double knockout alters presynaptic composition and ultrastructure. a** Schematic for simultaneous knockout of Liprin-α2 and -α3. Mice in which Liprin-α2 can be removed by cre recombination (Liprin-α2f/f) were generated and crossed to previously published constitutive Liprin-α3 knockout mice (Liprin-α3−/−, generated by CRISPR/Cas9-mediated genome editing, deleted sequence is represented with dashes)[19]. Cultured hippocampal neurons of Liprin-α2f/f/Liprin-α3−/− mice infected with lentivirus expressing cre recombinase (KOL23) were compared to neurons of Liprin-α2f/f x Liprin-α3+/− mice infected with lentiviruses that express an inactive cre recombinase (controlL23). **b, c** Example confocal images (**b**) and quantification (**c**) of neurons immunostained for Liprin-α2, Liprin-α3, RIM, Munc13-1, RIM-BP2, Bassoon, Ca_V2.1, PSD-95, Gephyrin, or GluA1 along with Synaptophysin (for Munc13-1, RIM-BP2, and Gephyrin) or Synapsin (all others) as vesicle marker, or Synapsin-1 and MAP2. Quantification in **c** was performed in regions of interest (ROIs) defined by the synaptic vesicle marker and normalized to the average controlL23 levels per culture, N = 30 images/ 3 independent cultures per genotype per protein of interest, except for Munc13-1 (N = 15/3), RIM-BP2 (N = 14/3), and Gephyrin (N = 14/3), p values: Liprin-α2, 2e-16 (***); Liprin-α3, 2e-16 (***); RIM, 3e-11 (***); Munc13-1, 3e-05 (***); RIM-BP2, 0.54; Bassoon, 0.07; PSD-95, 0.30; Gephyrin, 0.10; GluA1, 0.53; Ca_V2.1, 5e-10 (***); Synapsin, 7e-06 (***). **d–h** Example electron micrographs of synapses (**d**) and quantification (per section) of the number of synaptic vesicles (**e**), bouton size (**f**), PSD length (**g**), and number of docked vesicles (**h**) of neurons fixed by high-pressure freezing followed by freeze substitution, controlL23: N = 111 synapses/2 independent cultures, KOL23: N = 124/2, p values: **e**, 0.02 (*); **f**, 0.51; **g**, 0.95; **h**, 0.02 (*). All data were mean ± SEM. Significance was assessed using Mann–Whitney U tests, except for PSD-95 and Gephyrin in c, for which t-tests were used. All tests were two-sided. For synaptic localization of Liprin-α1 to -α4, see Supplementary Fig. 7, for generation of Liprin-α2f/f mice and analysis of Synaptophysin levels of the experiments shown in **c**, see Supplementary Fig. 8.

**Knockout of Liprin-α2 and Liprin-α3 alters presynaptic composition and ultrastructure in primary mouse hippocampal neurons**. If Liprin-α phase separation controls active zone assembly, Liprin-α knockout should impair active zone structure and function. We generated knockout mice to simultaneously ablate Liprin-α2 and Liprin-α3, the main synaptic Liprin-α proteins (Supplementary Fig. 7)[19,34]. Conditional Liprin-α2 knockout mice (Liprin-α2[f/f]), produced by homologous recombination with exon 14 flanked by loxP sites (Supplementary Fig. 8a–e), were crossed to homozygosity and subsequently bred to previously generated constitutive Liprin-α3 knockout mice (Liprin-α3[−/−])[19] (Fig. 4a). We used cultured hippocampal neurons of Liprin-α2[f/f]/Liprin-α3[−/−] mice infected with lentivirus expressing cre recombinase (to generate KO[L23] neurons) and neurons from Liprin-α2[f/f]/Liprin-α3[+/−] mice infected with lentiviruses that express truncated, inactive cre recombinase (to generate control[L23] neurons). First, we assessed synapse composition by measuring protein levels within synapses using confocal microscopy (Fig. 4b, c). Liprin-α2 and Liprin-α3 were efficiently removed and the remaining signals are typical for antibody background[22,40]. The levels of RIM and Munc13 were decreased by 25–35%, without significant changes in Bassoon, RIM-BP2, PSD-95, Gephyrin, and GluA1. The synaptic levels of $Ca_V2.1$ were increased by ~50%, as were those of Synapsin-1, but Synaptophysin levels were unchanged.

High-pressure freezing followed by freeze substitution and electron microscopic analyses was next used to investigate synaptic ultrastructure. The number of synaptic vesicles per profile was decreased by ~15% in KO[L23] synapses, without changes in the overall bouton size or postsynaptic densities (Fig. 4d–h). A ~25% reduction of docked vesicles (identified as vesicles with no detectable space between the electron-dense vesicular and target membranes) was observed upon Liprin-α2/α3 knockout, consistent with a partial loss of the docking proteins RIM and Munc13-1. We conclude that Liprin-α2 and Liprin-α3 are involved in maintaining presynaptic ultrastructure, specifically the number of vesicles per bouton and the number of docked vesicles.

We next asked using STED side-view analyses whether the changes in synaptic protein levels observed with confocal microscopy occur at active zones of both excitatory and inhibitory synapses (marked with PSD-95 and Gephyrin, respectively). This further circumvents confounds in ROI selection in the confocal experiments that may arise from modestly increased Synapsin levels and slightly decreased total synaptic vesicle numbers (Fig. 4b, c, e and Supplementary Fig. 8f, h). At KO[L23] synapses, Munc13-1, RIM, RIM-BP2, and $Ca_V2.1$ peaked at ~100 nm from the postsynaptic markers similar to control[L23] and in agreement with their active zone localization[19,41]. The levels of RIM-BP, PSD-95, Gephyrin, and Synaptophysin were unaffected. Decreased active zone levels of Munc13-1 and RIM were observed in excitatory and inhibitory KO[L23] synapses, while increased $Ca_V2.1$ levels were restricted to excitatory synapses (Fig. 5). This matches with the results obtained with confocal microscopy (Fig. 4b, c) and the electrophysiological phenotypes described below. We conclude that Liprin-α2 and -α3 are necessary for normal active zone structure.

**Synapse-specific impairments of neurotransmitter release after ablation of Liprin-α2 and -α3 in primary mouse hippocampal neurons**. The altered levels of Munc13-1, RIM, and $Ca_V2.1$ and the decreased docking predict changes in synaptic secretion. Indeed, in whole-cell electrophysiological recordings, the frequency of spontaneous miniature excitatory and inhibitory postsynaptic currents (mEPSCs and mIPSCs, respectively) was decreased in KO[L23] neurons, but their amplitudes were unchanged (Fig. 6a–c, 6j–l), indicating presynaptic roles for Liprin-α2 and -α3.

We used electrical stimulation or stimulation by hyperosmotic sucrose to evoke EPSCs (Fig. 6d–i) and IPSCs (Fig. 6m–r). Release evoked by an action potential is proportional to the product of the number of vesicles that can be released (readily releasable pool, RRP) and the likelihood of a vesicle to be released (vesicular release probability, $p$)[42,43]. For action-potential triggered EPSCs, NMDA-receptor currents were measured instead of AMPA-receptor currents to avoid network activity that is prominent in the cultured neurons when AMPA-receptors are not blocked. Similar to confocal and STED microscopy, we observed synapse-specific changes. At excitatory and inhibitory KO[L23] synapses, the RRP estimated by the application of hypertonic sucrose was decreased (Fig. 6h, i, q, r), quantitatively matching the reduction in Munc13, RIM, and docked vesicles (Figs. 4, 5). We estimated $p$ by measuring paired pulse ratios, where the response ratio of two consecutive pulses at short interstimulus intervals is inversely correlated with $p$[42]. Changes at excitatory KO[L23] synapses were indicative for increased $p$ (Fig. 6f, 6g), matching well with enhanced $Ca^{2+}$ channel levels (Figs. 4, 5). Together, the reduction in RRP and the increase in $p$ may offset one another. The evoked EPSC appeared unchanged, although there was a non-significant trend towards a decrease (Fig. 6d, e). In contrast, $p$ was unaffected at inhibitory synapses (Fig. 6o, p), matching with normal $Ca^{2+}$ channel levels, and leading to an overall decrease in the IPSC amplitude due to the RRP decrease (Fig. 6m, n). In summary, the electrophysiological phenotypes match with the structural active zone effects. Knockout of Liprin-α2/α3 leads to reduction in docking, protein machinery for docking and priming and the pool of releasable vesicles at excitatory and inhibitory synapses, and a select increase in $Ca^{2+}$ channels and release probability at excitatory synapses.

**S760 is necessary for presynaptic functions of Liprin-α3 in primary mouse hippocampal neurons**. We asked whether reexpression of wild type Liprin-α3 reverses the presynaptic phenotypes of KO[L23] neurons. Lentiviral expression of Liprin-α3 in KO[L23] neurons restored active zone levels of Liprin-α3 (Supplementary Fig. 9a–d), the RRP (Supplementary Fig. 9e, f) and the reduced active zone levels of RIM (Supplementary Fig. 9g, h). Hence, the active zone impairments at excitatory KO[L23] synapses are reversible by reexpression of Liprin-α3.

If Liprin-α3 functions depend on its propensity to phase separate, its ability to rescue should decrease when S760 is mutated to abolish phase separation. We directly compared the ability of wild type Liprin-α3 and Liprin-α3[SG] to rescue RRP and RIM levels (Fig. 6s–w). Liprin-α3 proteins (N-terminally tagged with an HA epitope) were expressed in KO[L23] neurons using lentiviral transduction. At 15 days in vitro (DIV15), both forms of Liprin-α3 were enriched at active zones, but the peak levels of Liprin-α3[SG] were somewhat decreased compared to Liprin-α3 (by ~15%; Supplementary Fig. 9i–l). Expression of wild type Liprin-α3 increased the RRP and RIM active zone levels by ~40% in KO[L23] neurons, while expression of Liprin-α3[SG] failed to produce a significant increase (Fig. 6t–w). We conclude that the PKC phosphorylation site of Liprin-α3, which drives phase separation, is important to restore the RRP and for normal active zone structure.

**PKC phosphorylation of Liprin-α3 at S760 acutely modulates active zone structure and function in primary mouse hippocampal neurons**. PLC/PKC signaling acutely enhances active zone assembly (Fig. 3) and neurotransmitter release[44–46]. We

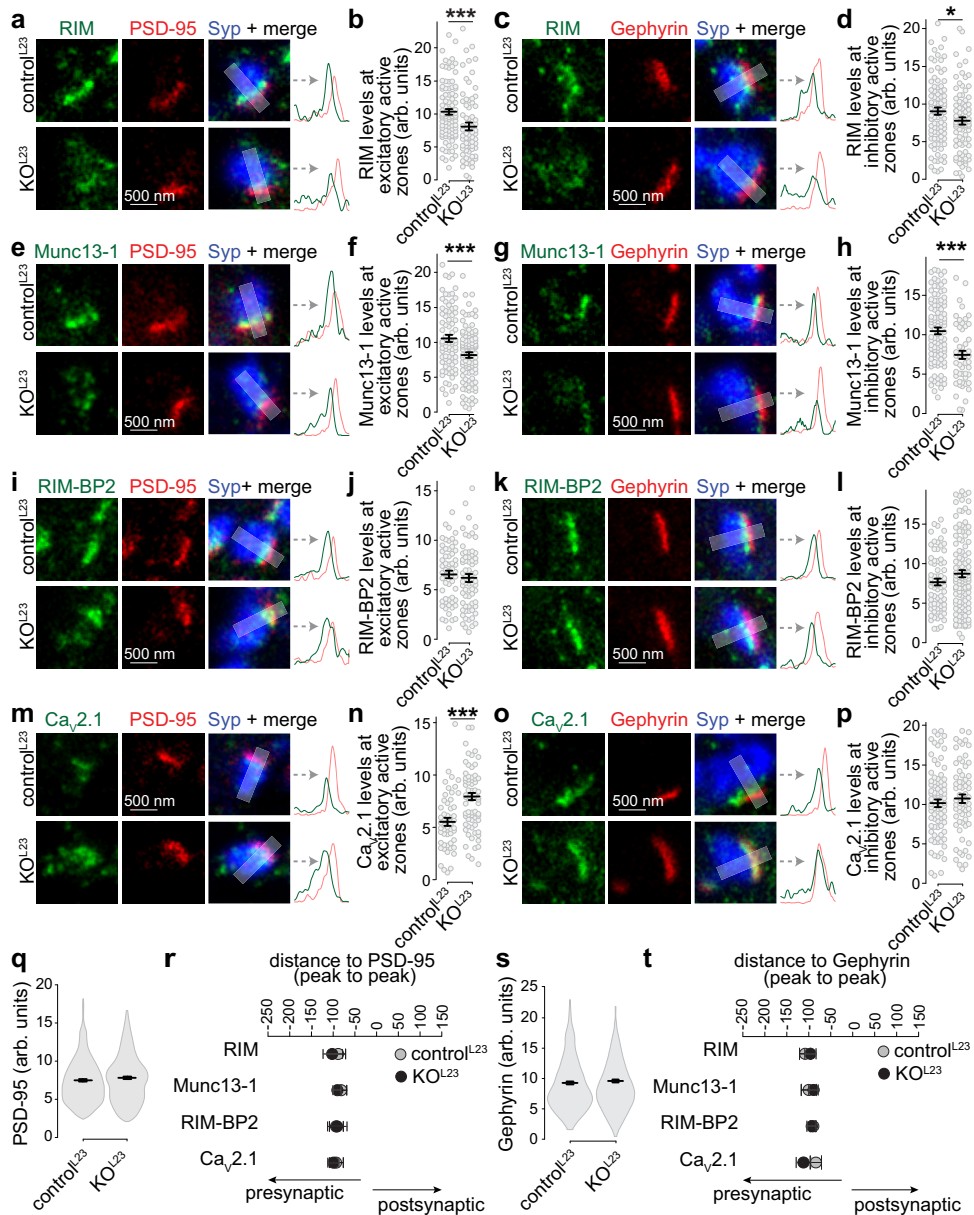

**Fig. 5 Active zone composition at excitatory and inhibitory synapses after Liprin-α2/α3 knockout. a–d** Example STED images and line profiles (**a, c**) and quantification of peak intensities (**b, d**) of RIM at Synaptophysin (Syp) positive excitatory side-view synapses identified by PSD-95 (**a, b**) or inhibitory side-view synapses identified by Gephyrin (**c, d**). **b**: control[L23]: $N = 96$ synapses/3 independent cultures, KO[L23]: $N = 69/3$, **d**: $N = 91/3$, KO[L23]: $N = 87/3$, $p$ values: **b**, 0.0008 (***); **d**, 0.04 (*). **e–p** Same as **a–d**, but for Munc13-1 (**e–h**), RIM-BP2 (**i–l**), and Ca$_V$2.1 (**m–p**). Munc13-1, **f**: control[L23]: $N = 79/3$, KO[L23]: $N = 99/3$, **h**: control[L23]: $N = 102/3$, KO[L23]: $N = 54/3$; RIM-BP2, **j**: control[L23]: $N = 57/3$, KO[L23]: $N = 66/3$, **l**: control[L23]: $N = 73/3$, KO[L23]: $N = 116/3$; Ca$_V$2.1, **n**: control[L23]: $N = 52/3$, KO[L23]: $N = 72/3$, **p**: control[L23]: $N = 87/3$, KO[L23]: $N = 66/3$, $p$ values: **f**, 0.0003 (***); **h**, 0.00005 (***); **j**, 0.49; **l**, 0.24; **n**, 0.0001 (***); **p**, 0.18. **q** Quantification of the peak intensity of PSD-95 in all line profiles, control[L23] $N = 286/3$, KO[L23] = 309/3, $p$ value: 0.21. **r** Quantification of the average distance of peaks of RIM, Munc13, RIM-BP2, and Ca$_V$2.1 to the peak of PSD-95, $N$ as in **b**, **f**, **j**, and **n**, $p$ values: RIM, 0.85; Munc13-1, 0.52; RIM-BP2, 0.86; Ca$_V$2.1, 0.67. **s** Quantification of the peak intensity of Gephyrin in all line profiles, control[L23] $N = 353/3$, KO[L23] = 323/3, $p$ value: 0.47. **t** Quantification of the average distance of peaks of RIM, Munc13, RIM-BP2, and Ca$_V$2.1 to the peak of Gephyrin, $N$ as in **d**, **h**, **l** and **p**, $p$ values: RIM, 0.98; Munc13-1, 0.84; RIM-BP2, 0.81; Ca$_V$2.1, 0.09. Data were shown as mean ± SEM. Significance was assessed by Mann–Whitney rank-sum tests (**b**, **d**, **h**, **l**, **q–t**) or $t$-tests (**f**, **j**, **n**, **p**). All tests were two-sided.

hypothesized that this enhancement may be mediated by phosphorylation and phase separation of Liprin-α3, and compared the effect of PKC activation by PMA in KO[L23] neurons expressing either wild type Liprin-α3 or Liprin-α3[SG], which does not form phase condensates. We note that despite their broad use to mimic activation of the PLC/PKC pathways[38,44–47], it remains uncertain how potentiation induced by phorbol esters relates to physiological synaptic plasticity. Neurons incubated with PMA for 20 min

showed enhanced mEPSC frequencies and amplitudes (Fig. 7a–f), as observed before[46], indicating that these pathways potentiate synaptic transmission through pre- and postsynaptic effectors. The mEPSC frequency increase was impaired by 50% in Liprin-α3[SG] expressing neurons (Fig. 7a–c), establishing that Liprin-α phosphorylation is important for this enhancement. Similarly, the RRP increase was significantly weakened when non-phosphorylatable Liprin-α3 was present (Fig. 7g–i). It is noteworthy that the RRP

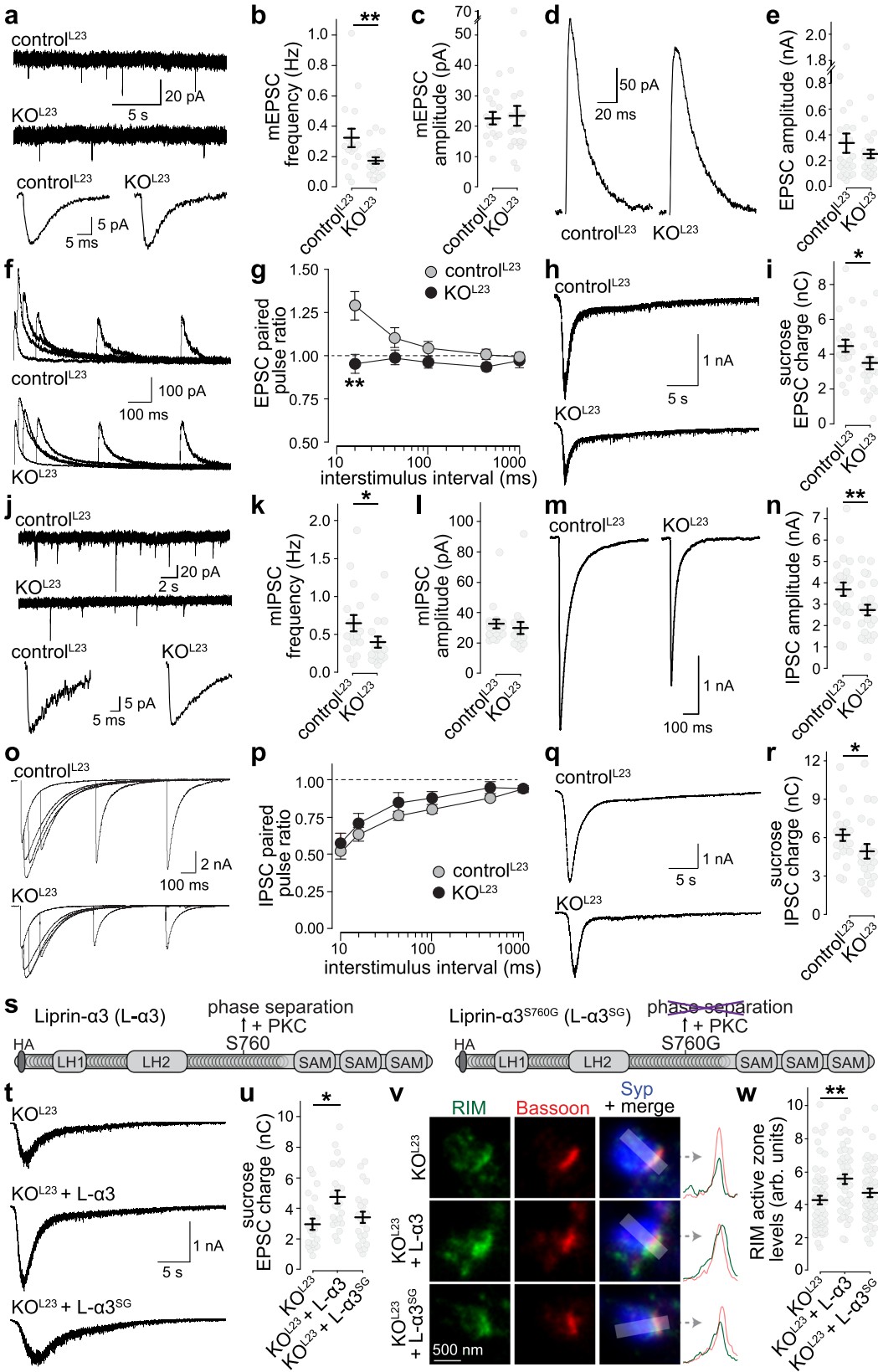

enhancement is likely overestimated because of the robust increase in mEPSC amplitude (Fig. 7f), and as a consequence the impairment in pool enhancement of Liprin-α3$^{SG}$ may be underestimated. In summary, these data indicate that PKC phosphorylation and phase separation of Liprin-α3 modulate the RRP.

We finally investigated whether Liprin-α3 phosphorylation and phase separation may control active zone structure. The lack of technology to selectively photobleach active zones within synapses with the necessary resolution of tens of nanometers prevents performing FRAP experiments to directly answer this

**Fig. 6 Liprin-α2/α3 double knockout impairs neurotransmitter release. a–c** Example traces (**a**) of spontaneous miniature excitatory postsynaptic current (mEPSC) recordings (top) and an averaged mEPSC of a single cell each (bottom), and quantification of mEPSC frequency (**b**) and amplitude (**c**), control[L23]: $N = 15$ cells/3 independent cultures, KO[L23]: $N = 21/3$, $p$ values: **b**, 0.003 (**); c, 0.55. **d, e** Example traces (**d**) and average amplitudes (**e**) of single action potential-evoked NMDA receptor-mediated EPSCs, control[L23]: $N = 25/3$, KO[L23]: $N = 26/3$, $p$ value: 0.63. **f, g** Example traces (**f**) and average NMDA-EPSC paired pulse ratios (**g**) at various interstimulus intervals, control[L23]: $N = 25/3$, KO[L23]: $N = 23/3$, $p$ values: genotype 0.00002, interstimulus interval 0.009, interaction 0.007, post-tests: 20 ms 0.002 (**), 50 ms 0.11, 100 ms 0.16, 500 ms 0.18, 1 s 0.24. **h, i** Example traces (**h**) and quantification (**i**) of the AMPA receptor-mediated EPSC charge in response to a local 10 s puff of 500 mOsm sucrose to estimate the RRP, control[L23]: $N = 21/3$, KO[L23]: $N = 25/3$, $p$ value: 0.045 (*). **j–r** Same as **a–i**, but for IPSCs, **j–l**: control[L23]: $N = 18/3$, KO[L23]: $N = 18/3$; **m + n**: control[L23]: $N = 23/3$, KO[L23]: $N = 24/3$; **o + p**: control[L23]: $N = 23/3$, KO[L23]: $N = 23/3$, **q + r**: control[L23]: $N = 22/3$, KO[L23]: $N = 22/3$, $p$ values. **k**, 0.046 (*); **l**, 0.12; **n**, 0.002 (**); **p**, genotype 0.37, interstimulus interval 2e-8, interaction 0.96; **r**, 0.011 (*). **s** Diagram of the rescue experiment with Liprin-α3 expression via lentiviral transduction. **t, u** Example traces (**t**) and quantification (**u**) of sucrose-triggered EPSCs, KO[L23]: $N = 22/3$, KO[L23] + Liprin-α3 (L-α3): $N = 23/3$, KO[L23] + Liprin-α3[S760G] (L-α3[SG]): $N = 23/3$, $p$ values (compared to KO[L23]): KO[L23] + L-α3, 0.006 (*); KO[L23] + L-α3[SG], 0.53. **v, w** Representative STED images (**v**) and quantification (**w**) of RIM at the active zone of side-view synapses as in Fig. 3i–j. KO[L23]: $N = 56$ synapses/3 independent cultures, KO[L23] + L-α3: $N = 43/3$, KO[L23] + L-α3[SG]: $N = 50/3$. $p$ values: KO[L23] + L-α3, 0.003 (**); KO[L23] + L-α3[SG], 0.18. All data were mean ± SEM. Significance was assessed using Mann–Whitney U- tests (**b, c, i, k, l**), t-tests (**e, n, r**), two-way ANOVA (**g, p**), or Kruskal–Wallis (**u, w**) with Tukey–Kramer (**g, p**) or Holm (**u, w**) post hoc comparison against KO[L23] (comparison between all groups were performed, but only results against KO[L23] are reported in **u** and **w**). All tests were two-sided. For a direct comparison of control[L23], KO[L23] and KO[L23] + L-α3 and for STED localization of rescue Liprin-α, see Supplementary Fig. 9.

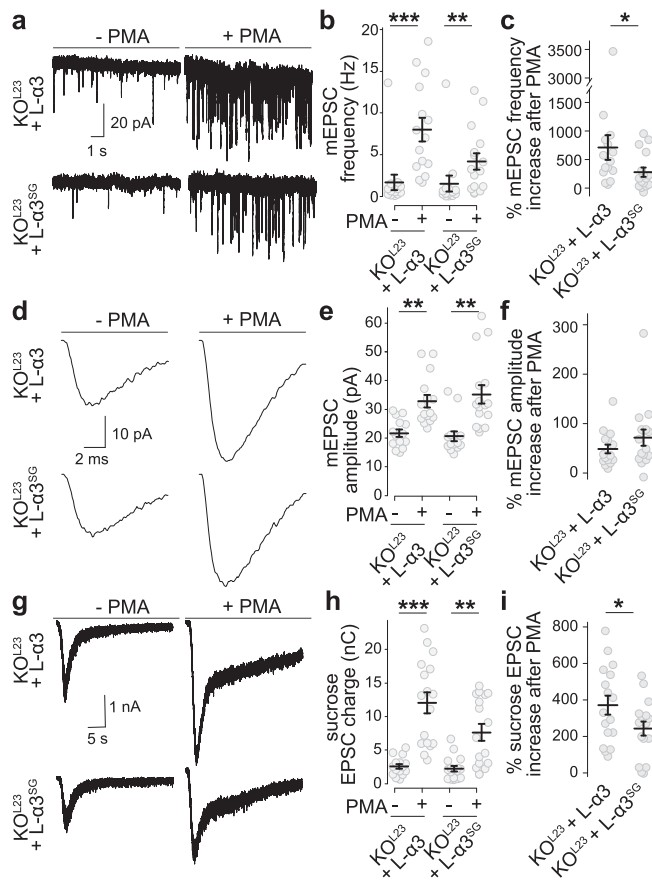

**Fig. 7 PKC phosphorylation of Liprin-α3 enhances synaptic vesicle release. a–c** Example traces (**a**) and quantification of mEPSC frequencies (**b, c**) in KO[L23] neurons rescued with wild type Liprin-α3 (L-α3) or non-phosphorylatable Liprin-α3 S760G (L-α3[SG]) that does not form phase condensates. The percent increase upon PMA addition over naïve conditions per culture is shown in **c**. KO[L23] + L-α3: $N = 14$ cells/3 independent cultures (−PMA) and 15/3 (+PMA); KO[L23] + L-α3[SG]: $N = 14/3$ (−PMA) and 16/3 (+PMA), $p$ values: **b**, KO[L23] + L-α3, 0.0004 (***), KO[L23] + L-α3[SG], 0.0084 (**); **c**, 0.03 (*). **d–f** Average mEPSCs from a single cell of each condition (**d**) and quantification of mEPSC amplitudes (**e, f**). $N$ as in **b, c**, $p$ values: **e**, KO[L23] + L-α3, 0.002 (**), KO[L23] + L-α3[SG], 0.001 (**); **f**, 0.15. **g–i** Example traces (**g**) and quantification (**h, i**) of the EPSC charge in response to a local 10 s puff of 500 mOsm sucrose to estimate the RRP. KO[L23] + L-α3: $N = 15/3$ (–PMA) and 17/3 (+PMA), KO[L23] + L-α3[SG]: $N = 17/3$ (–PMA) and 17/3 (+PMA), $p$ values: **h**, KO[L23] + L-α3, 0.00002 (***), KO[L23] + L-α3[SG], 0.002 (**); **i**, 0.03 (*). All data were mean ± SEM. Significance was assessed by Mann–Whitney rank sum test (**c, f, i**), or by Kruskal–Wallis test with Holm post hoc tests between all groups (**b, e, h**), and only results compared to the corresponding −PMA control are reported in **b, e**, and **h**. All tests were two-sided.

RIM. Together, these data support the model that active zone structure is rapidly modulated by PLC/PKC signaling via phosphorylation and phase separation of Liprin-α3 (Fig. 8j).

## Discussion

Recruitment of proteins into liquid phases enables the formation of membrane-less intracellular compartments[1,2]. We investigated molecular pathways that drive and modulate assembly of the presynaptic active zone. Our experiments reveal that (1) PKC phosphorylates Liprin-α3 at S760 to drive the formation of membrane-attached liquid condensates containing RIM1α and Munc13-1, (2) genetic ablation of the prominent synaptic Liprin-α proteins (Liprin-α2 and -α3) leads to defects in active zone structure and function, including reductions in RIM and Munc13-1, and (3) PKC phosphorylation of Liprin-α3 at S760 upregulates neurotransmitter release and active zone levels of RIM and Munc13-1. These results lead to a working model in which presynaptic phase separation triggered by Liprin-α3 phosphorylation rapidly modulates active zone structure (Fig. 8j).

S760 phosphorylation may lead to the formation of liquid condensates (1) by enhancing or enabling interactions of the phosphorylated linker, (2) by recruiting adapters, or (3) by

question. We instead assessed side-view synapses of KO[L23] neurons, or of KO[L23] neurons expressing either Liprin-α3 or Liprin-α3[SG]. In both rescue conditions, Liprin-α3, RIM, and Munc13-1 were enriched at the active zone. As observed in Fig. 3h–j, active zone levels of these proteins, but not of Bassoon, increased upon PMA addition by ~30–35% when Liprin-α3 was present (Fig. 8a–i and Supplementary Fig. 10). This increase, assessed either by quantifying raw fluorescence (Fig. 8b, e, h), or by normalization to the respective pre-PMA condition (Fig. 8c, f, i), was hampered at KO[L23] synapses or in those rescued with Liprin-α3[SG], with only ~10–15% enhancement of Munc13-1 and

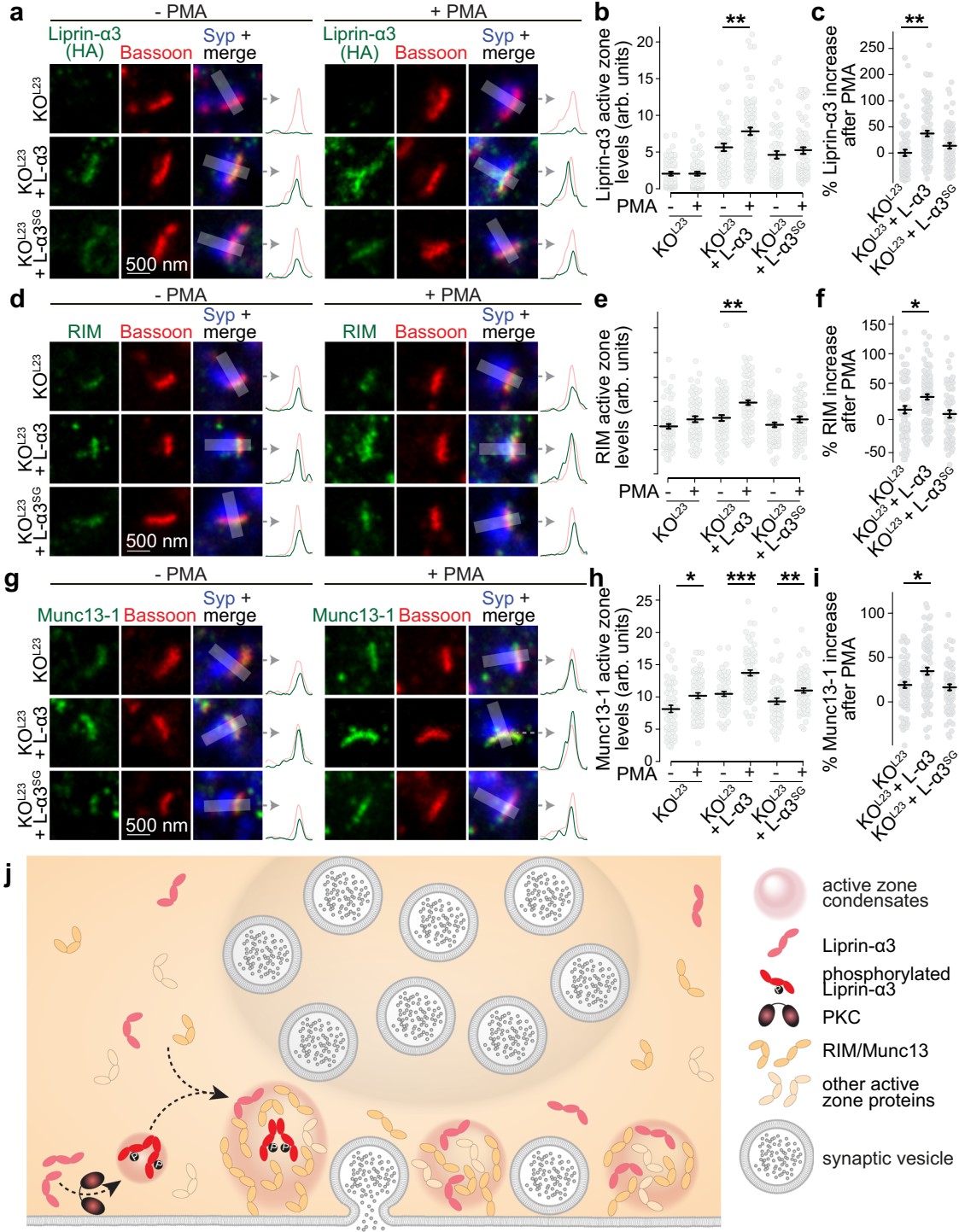

inducing Liprin-α3 conformational changes that expose previously occluded domains. These mechanisms may cooperate and could further function in phase-separation-independent ways. The third scenario appears likely because synaptic Liprin-α functions strongly depend on N-terminal sequences that mediate Liprin-α dimerization and active zone assembly[15–17]. S760 is not part of these sequences, but its phosphorylation could determine interactions and function of the N-terminus, for example through a previously proposed model in which intramolecular Liprin-α interactions mediate its functions[35]. This possibility is further supported by the observation that no single Liprin-α3 fragment was able to form condensates when expressed

alone, and sequences spanning multiple domains are necessary for Liprin-α3 phase separation.

There are notable differences in condensate formation across Liprin-α proteins. Only Liprin-α2 and Liprin-α3 form condensates in transfected cells. While Liprin-α3 condensates follow liquid dynamics, those formed by Liprin-α2 do not. Such biophysical properties may be related to different functions for these two proteins. For instance, roles in active zone remodeling may be unique to Liprin-α3 because other Liprin-α proteins are not subject to PKC-mediated triggering of phase separation. Baseline active zone assembly, however, is mediated by both Liprin-α2 and -α3 (this study and[19]), and this role may not need the

**Fig. 8 The S760 PKC phosphorylation site of Liprin-α3 acutely modulates active zone assembly. a–c** Example STED images and their intensity profiles (**a**) and quantification (**b**, **c**) of Liprin-α3 (detected by anti-HA antibodies) in side-view synapses in the presence or absence of PMA. Quantification of levels in arbitrary units is shown in **b**, and the increase upon PMA addition normalized to corresponding −PMA controls is shown in **c**, $KO^{L23}$: $N =$ 60 synapses/3 cultures (−PMA) and 71/3 (+PMA), $KO^{L23}$ + L-α3: $N = 54/3$ (−PMA) and 83/3 (+PMA); $KO^{L23}$ + L-α3$^{SG}$: $N = 63/3$ (−PMA) and 68/3 (+PMA), $p$ values: **b**, $KO^{L23}$, 0.91, $KO^{L23}$ + L-α3, 0.006 (**), $KO^{L23}$ + L-α3$^{SG}$, 0.42; **c** (compared to $KO^{L23}$), $KO^{L23}$ + L-α3, 0.002 (**), $KO^{L23}$ + L-α3$^{SG}$, 0.08. **d–i** Experiments as shown in **a–c**, but for RIM (**d–f**) and Munc13-1 (**g–i**), RIM (**d–f**): $KO^{L23}$: $N = 81/3$ (−PMA) and 61/3 (+PMA), $KO^{L23}$ + L-α3: $N =$ 75/3 (−PMA) and 84/3 (+PMA); $KO^{L23}$ + L-α3$^{SG}$: $N = 65/3$ (−PMA) and 59/3 (+PMA), Munc13-1 (**g–i**): $KO^{L23}$: $N = 54/3$ (−PMA) and 67/3 (+PMA), $KO^{L23}$ + L-α3: $N = 55/3$ (−PMA) and 67/3 (+PMA); $KO^{L23}$ + L-α3$^{SG}$: $N = 57/3$ (−PMA) and 62/3 (+PMA), $p$ values: **e**, $KO^{L23}$, 0.14, $KO^{L23}$ + L-α3, 0.00005 (***), $KO^{L23}$ + L-α3$^{SG}$, 1.0; **f** (compared to $KO^{L23}$), $KO^{L23}$ + L-α3, 0.02 (*), $KO^{L23}$ + L-α3$^{SG}$, 0.69; **h**, $KO^{L23}$, 0.01 (*), $KO^{L23}$ + L-α3, 0.00003 (***), $KO^{L23}$ + L-α3$^{SG}$, 0.004 (**); **i**, $KO^{L23}$ + L-α3, 0.03 (*), $KO^{L23}$ + L-α3$^{SG}$, 0.53. **j** Working model for the control of active zone structure through phase separation of Liprin-α3. The formation of phase condensates is triggered by PKC phosphorylation of Liprin-α3 at S760 and Munc13-1 and RIM are recruited into these release site condensates for boosting neurotransmitter secretion. Our data indicate that Liprin-α3 phosphorylation is not required for Liprin-α3 recruitment to active zone areas, but that Liprin-α3 phosphorylation followed by phase separation enhances the levels of Munc13 and RIM and boosts neurotransmitter release. Data were shown as mean ± SEM. Significance was assessed by Kruskal–Willis and Holm post hoc tests between all groups, and only results compared to the corresponding −PMA control (**b**, **e**, and **h**) or versus $KO^{L23}$ (**c**, **f**, and **i**) are reported. All tests were two-sided. For assessment of Bassoon levels, see Supplementary Fig. 10.

dynamics that result from liquid properties. Liprin-α1 and Liprin-α4 do not form condensates in transfected HEK293T cells and may operate though different mechanisms or lack important components for phase condensation in these cells. Importantly, Liprin-α1 and -α4 antibody-signals appear mostly non-synaptic and may be dendritic[34,48], and at least Liprin-α1 operates in neuronal arborization[49]. Finally, an independent, parallel recent study found that C.elegans Liprin-α/SYD-2 also forms phase condensates that mediate aspects of presynaptic assembly[50]. In summary, a picture emerges where phase separation and liquid properties may determine cellular functions of Liprin-α. One limitation in our gene knockout experiments is that Liprin-α1 and Liprin-α4 are present, and a full definition of vertebrate Liprin-α function will require new gene targeting experiments to remove these proteins as well.

A key question is how Liprin-α3 phases interact with other presynaptic phase condensates. RIM1α and RIM-BP2 form liquid condensates in vitro that may organize tethering of voltage-gated Ca²⁺ channels[6,51]. It is uncertain whether Liprin-α3 is part of the same phase within a nerve terminal, or whether multiple independent phases coexist. The current evidence is most compatible with a model of multiple distinct phases. First, active zone levels of RIM and Munc13-1 decrease upon ablation of Liprin-α2 and Liprin-α3, but those of Ca_V2.1 increase and those of RIM-BP2 are unchanged. Similarly, ablation of RIM-BP and RIM[52], or of RIM and ELKS[19,22], do not lead to loss of presynaptic Liprin-α. Hence, these proteins are likely in distinct protein complexes, or phases. Second, Ca_V2.1 and Munc13-1 do not co-localize when assessed with immunogold electron microscopy, suggesting the presence of separate clusters[53]. In aggregate, it appears most likely that distinct liquid assemblies exist within an active zone, one containing RIM1 and RIM-BP2 to tether Ca²⁺ channels[3,6], and a different phase with Liprin-α, RIM, and Munc13-1[54,55]. RIM may participate in multiple distinct active zone condensates, and how these partially overlapping phases acquire their unique molecular composition is an intriguing question. Liprin-α3 may be a key determinant, as Munc13-1 is more efficiently recruited to RIM condensates when Liprin-α3 is present, and Liprin-α3 can recruit either RIM or Munc13-1 on its own. While the recruitment of RIM by Liprin-α3 can be explained by known interactions[27], we are not aware of direct binding between Liprin-α3 and Munc13-1, and direct or indirect interactions may be involved. Nonetheless, the observation of reduced Munc13-1 levels after Liprin-α ablation in vertebrates fits well with a previous report that Liprin-α deletion resulted in decreased levels of a specific Munc13 protein at the fly neuromuscular junction[21].

Phase interactions may also be at play with synaptic vesicle clusters, which are organized via phase condensation through Synapsin and Synaptophysin[8,10]. Ablation of Liprin-α2 and -α3 mildly impaired vesicle clustering, and the levels of Synapsin were modestly increased, suggesting that there might be interplay between Liprin-α and vesicle phases. Release requires the transition of synaptic vesicles from the vesicle cluster to release sites. It is possible that RIM/Liprin-α/Munc13 phases embody sites that enable such transitions, as both RIM and Munc13 define secretory hotspots[54–56]. Liprin-α3, Munc13, and RIM may each be involved in the recruitment of vesicles from the vesicle phase to release sites, consistent with their roles in vesicle tethering or docking and their interactions with vesicular proteins[24,51,57–59]. As such, the docking reaction could be seen as the transition of a vesicle from Synapsin-phase association to active zone-phase association.

The existence of multiple phase separation-based pathways for active zone assembly may explain some of the difficulties in understanding its mechanisms. Removal of each protein family, for example RIM[51,60,61], RIM-BP[62,63], or Liprin-α2 and -α3 (this study), leads to at most partial assembly defects, but combinations of mutations are required to disrupt active zones[22,52,64]. This redundancy may also be the reason why active zone protein deletions can lead to synapse-specific secretory deficits[65,66]. In the case of Liprin-α, our work establishes that it is recruited to active zone condensates independently of phosphorylation (Fig. 3), and Liprin-α3$^{SG}$, which does not undergo PKC-induced phase separation, localizes to active zones (Fig. 5). Hence, PKC-mediated phase separation of Liprin-α3 is not required for its active zone targeting. Furthermore, RIM is targeted to the active zone mostly correctly upon Liprin-α2 and -α3 ablation. We conclude that Liprin-α2 and -α3, Liprin-α3 phosphorylation and its phase separation are not required for vertebrate active zone assembly. We propose that there are constitutive mechanisms for active zone assembly that do not necessitate Liprin-α3 phase separation. Instead, phase separation of Liprin-α3, likely after it has been anchored to the active zone, may modulate and strengthen presynaptic assembly through recruitment of additional active zone proteins.

While phosphorylation-mediated phase separation of Liprin-α3 may be partially active during development, it can be acutely engaged by activating PLC/PKC signaling in established synapses. Modulating active zone assemblies through this pathway is well suited to explain rapid changes during plasticity. We propose that phosphorylation of Liprin-α3 by PKC nucleates the transition of Liprin-α3 into liquid condensates to recruit RIM, Munc13-1, and

possibly other active zone proteins. This allows addition of secretory machinery to the membrane to enhance release (Fig. 8j). Modulation of Liprin-α3 phase separation by PKC complements other presynaptic mechanisms for PLC/PKC-triggered potentiation, including those mediated by Munc13[44], Munc18[45], and Synaptotagmin-1[46], further supporting the involvement of multiple parallel mechanisms[29].

## Methods

**Assessments of droplets in transfected HEK293T cells**. HEK293T cells were plated on 0.1 mm thick coverslips and transfected with plasmids expressing proteins of interest under the CMV promoter. About 500 ng of DNA per well (1.9 cm$^2$) were used for single plasmid transfections. If multiple plasmids were transfected, additional DNA was used at a 1:1 molar ratio. For assessment of fixed cells, fixation was performed using 4% paraformaldehyde 10–16 h after transfection. Drugs were added 15 min before cells were fixed unless otherwise noted, at the following concentrations: forskolin (10 μM, Sigma), PMA (1 μM, Sigma), caffeine (1 mM, Sigma), H-89 (5 μM, Abcam), bisindolylmaleimide-I (Bis-I, 0.1 μM, Sigma), KN-93 (1 μM, Abcam) and cells were fixed in the presence of drugs. For experiments including only PMA, the equivalent amount of DMSO was added to control. Images were acquired with a Leica SP8 Confocal/STED 3X microscope, using an oil-immersion 63X objective. For single protein expression, quantification was done manually, including only spherical condensates of >1 μm in diameter. To quantify the amount and size of protein structures produced by combined expression of Liprin-α3, RIM1α, and Munc13-1, or by combined expression of Liprin-α2 and Liprin-α3, the "Analyze particles" plug-in (Fiji) was used with automatic thresholding of the Munc13-1 or Liprin-α2 channel, respectively, and a minimum diameter of 1 μm. In all experiments comparing different proteins or treatments, the experimenter was blind to the condition throughout data acquisition and analyses.

For FRAP in live cells, HEK293T cells were plated on 35 mm plastic dishes containing 0.15 mm thick coverslips. Twelve to fifteen hours after transfection and 10 min after PMA addition, the dishes were transferred to the microscope stage and single droplets or peripheral condensates were photobleached using a 405 nm wavelength laser followed by image acquisition at a 1 (Fig. 1) or 3 (all other FRAP experiments) Hz sampling frequency in confocal mode. HEK293T cells were kept in the tissue culture medium containing 1 μM PMA and imaged at room temperature within 1 h of PMA addition. Regions of interest were drawn over prebleached structures and the percentage of intensity recovered was plotted as a function of time. $t_{1/2\ max\ recovery}$ was calculated as the time it takes for fluorescence to reach 50% of the maximum fluorescence recovered after bleaching. The baseline was assessed as the fluorescence intensity of the pre-bleached structure in one or the average of two images acquired right before bleaching. For fusion and fission analyses, droplet behavior was assessed 5 min after PMA addition during a time-lapse recording. Fusion was defined as two independent droplets (identified by their spherical shape and the dense fluorescence signal) that merged into a bigger condensate that remained stable for at least, 30 s. Conversely, fission was defined as an event in which a larger condensate generated two smaller ones. For quantification of relaxation, the decay of the aspect ratio (AR; longer over shorter condensate diameter) over time was fitted to:
$$AR(t) = AR\infty + (ARo - AR\infty)*exp(-t/\tau).$$
Confocal images were acquired using a Leica SP8 Confocal/STED 3X microscope with a 63X oil-immersion objective. The following N-terminally tagged (unless noted otherwise) plasmids were used: pCMV cerulean-Liprin-α3 (p471, rat), pCMV mVenus-Liprin-α3 (p472, rat), pCMV mVenus-Liprin-α3 Y648A+S650A+S651A (p516, rat), pCMV mVenus-Liprin-α3 S751A (p499, rat), pCMV mVenus-Liprin-α3 S760A (p500, rat), pCMV mVenus-Liprin-α3 S760G (p507, rat), pCMV mVenus-Liprin-α3 S760E (p503, rat), pCMV mVenus-Liprin-α3 S763A (p510, rat) and pCMV mVenus-Liprin-α3 S764A (p514, rat), pCMV mVenus-Liprin-α3-1-188 (p522, rat, amino acid numbering follows NM_001270985.2), pCMV mVenus-Liprin-α3-189-576 (p523, rat), pCMV mVenus-Liprin-α3-577-790 (p521, rat), pCMV mVenus-Liprin-α3-791-1192 (p524, rat), pCMV mVenus-Liprin-α1 (p920, mouse), pCMV mVenus-Liprin-α2 (p921, mouse), pCMV EGFP-Liprin-α4 (p466, mouse), pCMV cerulean (p46), pCMV mVenus (p51), pCMV RIM1α-mVenus (p587, rat, tag placed before the C2B domain, which does not interfere with protein function[40,51]), pcDNA Munc13-1-tdTomato (p888, rat, tag placed at the C-terminus), pCMV GFP-CamKIIα (p922, rat, Addgene #21226,[67]). Sequence analyses for the identification of IDRs were performed using IUPred2A (https://iupred2a.elte.hu/)[33] for mouse (UniProt ID: P60469), rat (Q91Z79), and human (O75145) Liprin-α3, and mouse Liprin-α1 (B2RXW8), -α2 (Q8BSS9), and -α4 (B8QI36).

**Expression and purification of GST-Liprin-α3 proteins**. GST-tagged fusion proteins were generated, expressed, and purified according to standard procedures[39]. Specifically, proteins were expressed at 20°C in E. coli BL21 cells after induction with 0.05 mM isopropyl b-D-1-thiogalactopyranoside for 20 h, and pelleted by centrifugation (45 min on 3500x$g$). For purification of GST-fusion proteins, bacterial pellets were resuspended and lysed for 30 min in PBS buffer supplemented with 0.5 mg/ml lysozyme, 0.5 mM EDTA, and a protease inhibitor

cocktail, followed by brief sonication and centrifugation (45 min at 11,200x$g$). Bacterial supernatants were incubated with glutathione-Sepharose resin (GE Healthcare) for 1.5 h at 4°C with gentle rotation, washed three times in PBS and stored until further use at 4°C (for no more than 5 days after purification). All steps after protein induction were conducted at 4°C and using ice-cold solutions. Protein concentrations were estimated by SDS-gel electrophoresis followed by Coomassie staining using BSA standards as a reference. The following GST-tagged proteins were produced from pGEX-KG2 constructs: pGEX Liprin-α3 1–188 (p567), pGEX Liprin-α3 189–576 (p568), pGEX Liprin-α3 577–790, (p566) and pGEX Liprin-α3 791–1192 (p570). Amino acid numbering follows NM_001270985.2.

**Assessment of Liprin-α3 phosphorylation**. For in vitro phosphorylation assays, 40 μg of the fusion protein bound to glutathione beads were incubated for 30 min in 200 μL of PKC reaction buffer (20 mM HEPES, 10 mM MgCl$_2$, 1.67 mM CaCl$_2$, 150 mM NaCl, 1 mM DTT) with 0.25 ng/μl PKC (Promega, V526A), 1 μM PMA, 1 μM phosphatidyl serine (Sigma, P7769), and 200 μM ATP (Sigma, A2383). For experiments in which phosphorylation was detected by autoradiography, 10 μCi $^{32}$P-γ-ATP (Perkin Elmer) were added to the PKC reaction mix and incubated for an additional 1 h at 30°C, followed by gel electrophoresis. For mass spectrometric analyses, the phosphorylated GST Liprin-α3 577–790 protein was isolated by SDS gel electrophoresis, Coomassie blue staining and cutting out of the protein band after the initial PKC reaction. The sample was processed by the HMS Taplin Mass Spectrometry Facility for identification of phosphorylated amino acid residues. For experiments using phosphospecific antibodies, samples were run on SDS-page gels and processed for Western blotting. For phosphorylation assays in HEK293T cells, the transfected cells were treated with PMA as described with or without pre-incubation with PKC inhibitors (Ro31-8220, 1 μm), run on SDS-page gels and processed for Western blotting.

**Generation of custom antibodies**. Custom antibodies (A231, A232, and 247) were produced following established protocols[41]. Phospho-specific Liprin-α3 antibodies were generated using keyhole lympet hemocyanin (KLH) conjugated CKAPKRK(pSer)IKSSIGR or CAPKRKSIKS(pSer)IGRL, for phospho-S760 and phospho-S764, respectively. For Liprin-α3 antibodies, GST-Liprin-α3-188-576 (p568, pGEX Liprin-α3-188-576) was expressed in and purified from BL21 bacteria. Peptides and GST-fusion proteins were injected into rabbits whose sera had been pre-screened to prevent nonspecific antibody signal. Rabbits were given boosters every 2 weeks and bleeds were collected every 3 weeks. Sera that showed the strongest bands and specificity in western blotting were affinity-purified[41].

**Western blotting**. Western blotting was performed following established protocols[39]. Samples were prepared in SDS sample buffer, run on SDS-PAGE gels and transferred to nitrocellulose membranes at 4°C for 6.5 h in buffer containing (per l) 200 ml methanol, 14 g glycine, and 6 g Tris, followed by 1 h blocking at room temperature in saline buffer with 10% nonfat milk powder and 5% normal goat serum. The membranes were incubated in primary antibodies overnight at 4°C in saline buffer with 5% milk and 2.5% goat serum, followed by 1 h incubation at room temperature with horseradish peroxidase-conjugated secondary antibodies prior to visualization of the protein band. Primary antibodies used: rabbit anit-Liprin-α2 (A13, 1:500) and -Liprin-α3 (A115, 1:500) were gifts from S. Schoch[34]; rabbit anti-phospho-S760 Liprin-α3 (generated for this study; A231; 1:1000) and anti-phospho-S764 Liprin-α3 (generated for this study; A247 and A248; 1:1000); mouse anti-HA (A12, 1:500; RRID: AB_2565006); mouse anti-Synaptophysin (A100, 1: 5000; RRID:AB_887824) and mouse anti-Synapsin-1 (A57, 1:5000; RRID: AB_2617071). Secondary antibodies used: goat anti-rabbit HRP conjugated (S53, 1:10000, RRID: AB_2334589, 1:10000), goat anti-mouse HRP conjugated (S52, 1:10000, RRID: AB_2334540). Three 5 min washes were performed between steps.

**Neuronal cultures and production of lentiviruses**. Primary hippocampal cultures were prepared following established protocols[19,39–41]. Newborn (P0–P1) pups were anesthetized on ice slurry prior to hippocampal dissection. Hippocampi were digested and dissociated, and neurons were plated onto glass coverslips in plating medium composed of Minimum Essential Medium (MEM) supplemented with 0.5% glucose, 0.02% NaHCO3, 0.1 mg/ml transferrin, 10% Fetal Select bovine serum, 2 mM L-glutamine, and 25 mg/ml insulin. Twenty-four hours after plating, plating medium was exchanged with growth medium composed of MEM with 0.5% glucose, 0.02% NaHCO3, 0.1 mg/ml transferrin, 5% Fetal Select bovine serum (Atlas Biologicals FS-0500-AD), 2% B-27 supplement, and 0.5 mM L-glutamine. At DIV2–3, 4 mM Cytosine b-D-arabinofuranoside (AraC) was added. Cultures were kept in a 37°C tissue culture incubator until DIV15–17. Lentiviruses were produced in HEK293T cells maintained in DMEM supplemented with 10% fetal bovine serum and 1% penicillin/streptomycin. HEK293T cells were transfected using the Ca$^{2+}$ phosphate method with the lentiviral packaging plasmids REV, RRE, and VSV-G and a separate plasmid encoding the protein of interest, at a molar ratio 1:1:1:1. Twenty-four hours after transfection, the medium was exchanged to neuronal growth medium and, 18–30 h later the supernatant was used for immediate transduction. Neuronal cultures were infected at DIV4–5 days after plating with lentiviruses expressing GFP-Cre or an inactive variant of GFP-

Cre expressed under the human Synapsin promotor[68]. For rescue, cultures were infected at DIV1–2 with a lentivirus expressing Liprin-α3 or Liprin-α3[SG], or an empty lentivirus as control. pFSW HA-Liprin-α3 S760G was generated for this study; pFSW control (p008) and pFSW HA-Liprin-α3 (p526) were previously described[19]. For PMA experiments, PMA was added 15–20 min before fixation to a final dilution of 1 μM (from a 1 mM stock diluted in DMSO), and neurons were washed and fixed or recorded in the presence of the drug. "−PMA" controls were incubated in the same amount of DMSO.

**Immunofluorescence staining and confocal microscopy of neurons.** Neurons grown on #1.5 (for STED) or #1.0 (confocal) glass coverslips were fixed in 4% paraformaldehyde (or 2% for Ca$_V$2.1 staining) for 10 min at DIV15–17, blocked and permeabilized in blocking solution (3% BSA/0.1% Triton X-100/PBS) for 1 h, incubated overnight with primary antibodies followed by overnight incubation with Alexa-conjugated secondaries, and mounted onto glass slides. Antibodies were diluted in blocking solution. For STED imaging, coverslips were additionally post-fixed in 4% paraformaldehyde for 10 min. Three 5 min washes with PBS were performed between steps. All steps were performed at room temperature except for antibody incubations (4 °C). Primary antibodies used: mouse anti-Bassoon (A85, 1:50; RRID:AB_11181058), rabbit anti-Liprin-α1 (A121; 1:100; gift from S. Schoch[34]), -α2 (A13, 1:250; gift from S. Schoch[34]), -α3 (A115, 1:250; gift from S. Schoch[34]), and -α4 (A2, 1:100; gift from S. Schoch[34]), rabbit anti-RIM (A58, 1:500; RRID: AB_887774), mouse anti-PSD-95 (A149, 1:500; RRID: AB_10698024), mouse anti-Gephyrin (A8, 1:500; RRID:AB_2232546), mouse anti-Synapsin-1 (A57, 1:500; RRID: AB_2617071), guinea pig anti-Synaptophysin (A106, 1:500; RRID: AB_1210382), rabbit anti-RIM-BP2 (A126, 1:500; RRID: AB_2619739), rabbit anti-Munc13-1 (A72, 1:500; RRID: AB_887733), rabbit anti-Ca$_V$2.1 (A46, 1:500; RRID: AB_2619841), mouse anti-HA (A12, 1:500; RRID: AB_2565006), rabbit anti-MAP2 (A139, 1:500; RRID: AB_2138183), mouse anti-MAP2 (A108, 1:500; RRID: AB_477193), mouse anti-GluA1 (A82; 1:100; RRID:AB_2113443), and rabbit anti-Liprin-α3 (A232, 1:500, custom-made). Secondary antibodies used: goat anti-rabbit Alexa Fluor 488 (S5; 1:250, RRID:AB_2576217), goat anti-mouse IgG1 Alexa Fluor 488 (S7, 1:250, RRID: AB_2535764), goat anti-mouse IgG1 Alexa Fluor 555 (S19, 1:2500, RRID: AB_2535769), goat anti-mouse IgG2a Alexa Fluor 555 (S20, 1:250, RRID: AB_2535776), goat anti-guinea pig Alexa Fluor 633 (S34, 1:500, RRID: AB_2535757), and goat anti-mouse Alexa Fluor 633 (S32, 1:500, RRID: AB_2535718). Confocal images were taken on an Olympus FV1200 confocal microscope equipped with a 60X oil immersion objective or a Leica SP8 Confocal/STED 3X microscope with a 63X oil immersion objective. Images of experiments with multiple groups were acquired within a single session per culture and identical settings for each condition were used within an imaging session. For quantitative analyses of synaptic protein levels, the synaptic vesicle marker signal was used to define puncta as ROIs, and the average intensity within ROIs was quantified after local background was subtracted using the "rolling average" ImageJ plugin (diameter = 1.4 μm). Data were plotted normalized to the average intensity of the control group (control[L23]) per culture. For co-localization analyses, the "Coloc 2" ImageJ plugin was used with default thresholding. For example images in figures, brightness and contrast were linearly adjusted equally between groups and images were interpolated to meet publication standards. All data were acquired and analyzed by an experimenter blind to genotype and/or condition.

**STED imaging of synapses.** STED microscopy was performed following established procedures[19,39–41]. Images were acquired with a Leica SP8 Confocal/STED 3X microscope equipped with an oil-immersion 100×1.44-N.A objective, white lasers, STED gated detectors, and 592 and 660 nm depletion lasers. Synapse-rich areas were selected and were scanned at 22.5 nm per pixel. Triple color sequential confocal scans were followed by dual-color sequential STED scans. Identical settings were applied to all samples within an experiment. For quantification, side-view synapses (selected while blind to the protein of interest) were defined as synapses that contained a vesicle cluster (imaged in confocal mode, >300 nm wide) with an elongated Bassoon, Gephyrin, or PSD-95 (active zone or postsynaptic density markers, respectively, imaged by STED) structure along the edge of the vesicle cluster[19,39–41]. A 1 μm-long, 250-nm-wide profile was selected perpendicular to the active zone/postsynaptic density marker and across its center. The intensity profile was then obtained for markers and for the protein of interest. Peak levels of the protein of interest were measured as the maximum intensity of the line profile within 100 nm of the active zone/postsynaptic density marker peaks (estimated active zone area based on ref. [19]) after applying a five-pixel rolled average. For analyzing overall Bassoon densities and areas in STED images, we used a custom-written code that we previously described and used for the same purpose[41] performing automatic two-dimensional segmentation with a size filter of 0.04–0.4 μm², and without considering the shape or orientation (available at https://github.com/kaeserlab/3DSIM_Analysis_CL and https://github.com/hmslcl/3D_SIM_analysis_HMS_Kaeser-lab_CL). Only for representative images, a smooth filter was added and brightness and contrast were linearly adjusted using ImageJ. Equal adjustments were performed for all images within a given experiment. Finally, images were interpolated to match publication standards. Quantitative analyses were performed on original images without any processing, and all data were acquired and analyzed by an experimenter blind to genotype and/or

condition. For generating figures, the measured arbitrary STED intensity values were divided or multiplied by an arbitrary factor for displaying the data in a suitable range. The factor varied from ~1 to ~20 and was identical within each data set of an experiment, but different for different data sets. The measured arbitrary values before division or multiplication are provided in the source data file. For PMA experiments, neurons were preincubated for a total of 20 min before fixation with PMA (for the "+ PMA" condition, at a final concentration of 1 μM PMA diluted from a 1 mM stock in DMSO into the culture medium) or in the corresponding amount of DMSO (for the "−PMA" condition). To calculate the "% increase after PMA", the data were normalized to the average of the corresponding −PMA condition in each culture.

**Mouse lines.** Liprin-α2 (Ppfia2) mutant mice were acquired from MRC Harwell (C57BL/6N-Ppfia2 < tm1a(EUCOMM)Hmgu > /H, HEPD0651_9_D04, EM:09631)[69]. The mice were generated by homologous recombination and first were crossed to Flp-expressing mice[70] to remove the LacZ/Neomycin cassette to generate the conditional allele, which contains loxP sites flanking exon 14. Conditional Liprin-α2 mice were kept as homozygotes, and genotyped using oligo-nucleotide primers GCCTCTTAACATTCACTGTACC and CCAGTGTGTACTG GAGACAAGC for the wild-type allele (336 band), and GGCTCTTAACATTC ACTGTACC and CTGCGACTATAGAGATATCAACC for the floxed allele (517 band) (Supplementary Table 1). To generate Liprin-α2/α3 double mutant mice, conditional Liprin-α2 knockout mice were crossed to previously described constitutive Liprin-α3 mice that were generated by CRISPR/Cas9-mediated genome editing[19]. The line was maintained using intercrosses between Liprin-α2$^{f/f}$/Liprin-α3$^{−/−}$ and Liprin-α2$^{f/f}$/Liprin-α3$^{+/−}$ mice. For experiments, hippocampal neurons cultured from individual P0 Liprin-α2$^{f/f}$/Liprin-α3$^{−/−}$ pups were infected with lentivirus expressing Cre recombinase (to generate KO[L23] neurons) and compared to Liprin-α2$^{f/f}$ x Liprin-α3$^{+/−}$ littermates infected with lentiviruses that express a truncated, inactive mutant of Cre (to generate control[L23] neurons)[68], both expressed via a human Synapsin promoter. For rescue experiments comparing Liprin-α3 with Liprin-α3$^{SG}$, the genotype of the breeders was Liprin-α2$^{f/f}$/Liprin-α3$^{−/−}$ and neurons were cultured from pooled hippocampi from multiple pups of the same litter, followed by addition of rescue virus and of lentivirus expressing Cre recombinase as described under the neuronal culture protocol. Mice were housed in a mouse facility room at 20–24 °C (set point 22 °C) and 35–70% (set point 50%) humidity on a regular dark/light cycle. All animal experiments were approved by the Harvard University Animal Care and Use Committee.

**Electron microscopy of cultured neurons.** Electron microscopy was performed following established protocols[22,41]. Neurons grown on 6 mm sapphire coverslips were transferred to extracellular solution containing (in mM) 140 NaCl, 5 KCl, 2 CaCl$_2$, 2 MgCl$_2$, 10 glucose, 10 Hepes (pH 7.4, ~310 mOsm), and 50 μM D-AP5, 50 μM picrotoxin and 20 μM CNQX, and subsequently frozen with a Leica EM ICE high-pressure freezer at DIV15–17. After freeze substitution (in acetone containing 1% osmium tetroxide, 1% glutaraldehyde, and 1% H$_2$O), samples were embedded in epoxy resin and sectioned at 50 nm with a Leica EM UC7 ultra-microtome. Samples were imaged with a JEOL 1200EX transmission electron microscope equipped with an AMT 2k CCD camera. Images were analyzed using SynapseEM, a MATLAB macro provided by Dr. Matthijs Verhage. Bouton size was calculated from the perimeter of each synapse. Docked vesicles were defined as vesicles touching the presynaptic plasma membrane (with no space between the electron-dense vesicular and target membranes) apposed to the PSD. All data were acquired and analyzed by an experimenter blind to the genotype.

**Correlative light-electron microscopy.** HEK293T cells were grown on photo etched gridded coverslip and fixed 12–16 h after transfection in 2.5% glutaraldehyde, 2% sucrose, 50 mM KCl, 2.5 mM MgCl$_2$, 2.5 mM CaCl$_2$, and 50 mM cacodylate (pH 7.4) for 2 h at 4 °C. A spinning disk confocal microscope (3i, Denver, Colorado) equipped with an oil-immersion 63×1.4 N.A. objective and 488/ 561 nm lasers was used for the acquisition of fluorescent images. After image acquisition, samples were stained for 2 h in staining solution I (SSI; consisting of 1% OsO4, 1.25% potassium hexacyanoferrate in 100 mM PIPES, pH 7.4,), followed by staining solution II (prepared by diluting 100 times SSI in 1% tannic acid) for 30 min, and incubated in 1% uranyl acetate overnight. All staining steps were done on ice, the sample was protected from light, and three 5-min washes with ice-cold milli-Q water were performed between steps. Samples were dehydrated with increasing ethanol concentrations (30, 50, 70, 90, and 100%), followed by two washes in 100% acetone, embedded in epoxy resin, baked at 60 °C for at least 36 h and sectioned at 50 nm with a Leica EM UC7 ultramicrotome. A JEOL 1200EX transmission electron microscope equipped with an AMT 2k CCD camera was used for image acquisition. Fluorescent and electron microscopy images were aligned with the BigWarp plugin (ImageJ) using the electron micrograph as a fixed image and different arbitrary references for alignment. As references, cell features such as the nucleus and the plasma membrane and the fluorescent signals were used. Multiple independent alignments using different references were conducted to confirm correct alignment.

**Electrophysiology.** Electrophysiological recordings were performed following previously used procedures[22,41]. DIV15–16 neurons were recorded in whole-cell patch-clamp configuration at room temperature in extracellular solution containing (in mM) 140 NaCl, 5 KCl, 1.5 CaCl$_2$, 2 MgCl$_2$, 10 HEPES (pH 7.4) and 10 Glucose. Glass pipettes were pulled at 2–4 MΩ and filled with intracellular solutions containing (in mM) 120 Cs-methanesulfonate, 10 EGTA, 2 MgCl$_2$, 10 HEPES-CsOH (pH 7.4), 4 Na$_2$-ATP, and 1 Na-GTP for excitatory transmission; and 40 CsCl, 90 K-Gluconate, 1.8 NaCl, 1.7 MgCl$_2$, 3.5 KCl, 0.05 EGTA, 10 HEPES, 2 MgATP, 0.4 Na$_2$-GTP, 10 phosphocreatine, CsOH (pH 7.4) for inhibitory transmission. For evoked responses, 4 mM QX314-Cl was added to the intracellular solution to block sodium channels. Neurons were clamped at −70 mV for IPSC and AMPAR-EPSC recordings, or +40 mV for NMDA-EPSCs. Series resistance was compensated to 5–5.5 MΩ. Recordings in which the series resistance increased to >15 MΩ before compensation were discarded. mEPSCs, mIPSCs, and sucrose-evoked release were measured in extracellular solution supplemented with 1 μM TTX, 50 μM D-AP5, and either 50 μM picrotoxin (for EPSCs) or 20 μM CNQX (for IPSCs). 500 mM hypertonic sucrose was applied for 10 s, and the integral of the first 10 s of the response was used to estimate the RRP. For PMA experiments, neurons were preincubated with PMA for the "+PMA" condition (at a final concentration of 1 μM, diluted from a 1 mM stock in DMSO into the recording solution) or in the corresponding amount of DMSO for the "−PMA" condition for 20 min before starting recording, and a maximum of two cells per coverslip were recorded within a total of 40 min after drug addition (20 min of preincubation, 20 min of recording). To calculate the "% increase after PMA", the data were normalized to the average of the corresponding −PMA condition in each culture. Action potential-evoked responses were elicited by focal bipolar electrical stimulation with an electrode made from Nichrome wire and recorded in extracellular solution supplemented with 20 μM CNQX and either 50 μM D-AP5 (for IPSCs) or 20 mM PTX (for NMDAR-EPSCs). A Multiclamp 700B amplifier and a Digidata 1550 digitizer were used for data acquisition, sampling at 10 kHz and filtering at 2 kHz. Data were analyzed using pClamp. In all experiments, the experimenter was blind to the condition throughout data acquisition and analyses.

**Statistics.** Normality and homogeneity of variances were assessed using Shapiro or Levene's tests, respectively. When test assumptions were met, parametric tests (t-test or one-way ANOVA) were used. Otherwise, the non-parametric tests (Mann–Whitney U or Kruskal–Wallis) were used. For paired pulse ratios, a two-way ANOVA was used. Tukey–Kramer or Holm corrections for multiple testing were applied for parametric and nonparametric post hoc testing. Chi-square tests were used to assess mouse survival ratios. All data were analyzed by an experimenter blind to the drug condition and/or genotype. For each dataset, the specific tests used are stated in the figure legends. Data were plotted as mean ± SEM, accompanied by scatterplots showing all individual data points (for experiments with <150 data points) or smoothed violin plots (for experiments with >150 data points), significance in figures is shown at $p < 0.05$ (*), $p < 0.01$ (**), or $p < 0.001$ (***). Exact $p$ values are provided in each figure legend.

**Reporting summary.** Further information on research design is available in the Nature Research Reporting Summary linked to this article.

## Data availability
Data sets presented in this study are included in full whenever possible, including display of individual data points. Data were available from the corresponding author upon reasonable request. Biological materials including mutant mice and custom antibodies generated for this study will be shared upon request within the limits of respective material transfer agreements for as long as they are available in the laboratory. Source data are provided with this paper.

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

## Acknowledgements

We thank J. Wang, E. Atwater, M. Sanghvi, and M. Han for technical support, Drs. R. Held and C. Tan for help and advice, and all members of the Kaeser laboratory for insightful discussions. We thank Dr. S. Schoch for Liprin-α antibodies, and Drs. M. Verhage and J. Broeke for the SynapseEM MATLAB macro. This work was supported by grants from the NIH (R01NS083898 and R01MH113349 to P.S.K., R35GM130386 to T.K.), the Lefler Foundation (to P.S.K.), the Armenise Harvard Foundation (to P.S.K.), a grant from the Novo Nordisk Foundation/Danish Technical University (NNF16OC0022166 to T.K.), a Biogen Sponsored Research Agreement (to T.K), and fellowships from the Alice and Joseph E. Brooks postdoctoral fund (to J.E.-M.), the Croucher foundation (to M.Y.W.), Lefler foundation (to M.Y.W.), and the NSF (graduate research fellowship DGE1144152 to S.S.H.W.). We thank the Sanger Institute for generating the *Ppfia2* ES cells and the Mary Lyon Centre, as part of MRC Harwell, for generating the mutant mice. We acknowledge the Neurobiology Imaging Facility (supported by a P30 Core Center Grant NS072030), the Taplin Mass Spectrometry Facility, and the Electron Microscopy Facility at Harvard Medical School.

## Author contributions

Conceptualization, J.E.-M., M.Y.W., and P.S.K.; Methodology, J.E.-M., M.Y.W., G.d.N., H.N., and T.K.; Investigation, J.E.-M., M.Y.W., S.S.H.W., H.N., and G.d.N.; Formal analysis, J.E.-M., M.Y.W., S.S.H.W, G.d.N., T.K., and P.S.K.; Writing-original draft, J.E.-M., and P.S.K.; Writing-review and editing, J.E.-M, G.d.N., H.N., T.K., and P.S.K.; Supervision, P.S.K.; Funding acquisition P.S.K.

## Competing interests

S.S.H.W. is currently an employee of RA Capital Management LP. M.Y.W. is currently and employee of Novartis. The remaining authors declare no competing interests.
