## [Peer Review File · Nature Communications]

Reviewers' Comments:

Reviewer #1:

Remarks to the Author:

Accumulating evidence indicate that phase separation is a general mechanism for organizing compartmentalized molecular machineries both in pre- and post-synaptic sides of synapses. In the presynaptic active zones, RIM and RIM-BP have been shown to undergo phase separation in vitro; Munc13-1 has been observed to form supramolecular nano-assembly in living neurons and non-neuronal cells; ELKS has been detected as cytoplasmic clusters with properties of liquid condensates. In this manuscript, Emperador-Melero et al. report that Liprin- α 3 undergoes phase separation in a PKC phosphorylation-dependent manner. Interestingly, of the four Liprin- α isoforms, only the two that are enriched in presynaptic active zones (α 2 and α 3) undergo phase separation, with α 2 constitutively forming droplets in HEK293T cells and α 3 forming condensates only after PKC-mediated phosphorylation on a single Ser residue. Using CLEM, the authors discovered that RIM and Munc13-1, can form colocalized condensates with Liprin- α 3 beneath the plasma membranes in HEK293T cells, recapitulating the active zone-like structure in synapses. The authors generated Liprin- α 2/ α 3 double knockout mice and performed elegant super-resolution imaging and electrophysiological studies that provide convincing results showing essential roles of Liprin- α 3 and its phosphorylation by PKC in organizing the active zone structure and in controlling neurotransmitter releases in neurons.

This is a comprehensive and high quality study covering boarded areas including biochemistry, cell biology, and electrophysiology. Most importantly, the authors have been able to demonstrate that formation of protein condensates (using Liprin- α 3 as the example) is critical for the formation and physiological function of active zones in presynaptic boutons. Therefore, the study is very timely and much desired in the field of phase separation in neuronal synapses specifically and in cell biology in general. The study is well designed. Experiments are performed with high quality, and results are clearly described and presented. This reviewer recommends highly of this work for publication in Nature Communication. I do have a few minor issues for the authors to consider:

1. Figure 1a: the authors used inhibitors and activators of PKA, PKC and CaMKII, and only PKC activation led to Liprin- α 3 droplets formation. But the expression of CaMKII in HEK293T cells is low, so it's perhaps not surprising that the cells do not respond to CaMKII activator or inhibitor. A possible experiment to test potential impact of CaMKII phosphorylation is to co-transfect Liprin- α 3 and CaMKII in HEK293T cells. As suggested by the Extended data figure 2b&2c, S764 phosphorylation also contributes to Liprin- α 3 phase separation, implying other kinase(s) may participate in the regulation of Liprin- α 3 phase separation.
2. Extended data figure 2f: both Liprin- α 2 and Liprin- α 3 (+PMA) forms "ring-like" structures instead of "droplet-like" structure shown in other figures. What are these "ring-like" structures?
3. Extended data figure 3a: Munc13-1 directly interacts with RIM but is not known to bind to Liprin- α 3. Curiously, Munc13-1 formed co-localized patches with Liprin- α 3 but not with the RIM droplets. Additionally, co-transfection of Munc13-1 with Liprin- α 3 resulted in very striking morphology changes of the Liprin- α 3 condensates. Maybe the authors can comment on this observation that is rather counterintuitive.
4. Figure 6: is there any reason why the KOL23 control is not included in the figure.

Reviewer #2:

Remarks to the Author:

The last years have seen the emergence of liquid-liquid phase separation (LLPS) as a new concept to organize membrane-less compartments. Both assembly but also dynamic regulation of the presynaptic active zone scaffold might well be organized by LLPS. Indeed, a previous study suggested that RIM, RIM-BP as well as the VGCC C-term might engage into reversible liquid-liquid phase separation. However, direct genetic evidence to document the regulatory role of LLPS is missing and not necessarily easy to establish, so that the question whether LLPS indeed is a regulatory principle still pertains.

In this study, the Kaeser lab investigates active zone scaffold proteins in the LLPS context in HEK cells and cultivated hippocampal neurons. In HEK cells, PMA application (PLC-induced generation of diacylglycerol and activates PKC) provoked the reversible formation of Liprin- α 3 spherical

condensates indicative of LLPS. FRAP analysis demonstrated liquid like features of the Liprin- α 3 condensates. Again, using HEK cells, they find that PKC phosphorylation of serine-760 (S760) of Liprin- α 3 triggers Liprin α 3 LLPS "in vivo". S760A 136 and S764A Liprin- α 3 mutations interfered with PMA-induced droplet formation in HEK cells. The defect in cellular phosphorylation via anti-phospho-S760/764 antibodies.

They then turn to cultured hippocampal neurons and assess active zone levels of endogenous Liprin- α 3, RIM and Munc13-1 at synapses using STED microscopy. They show that PMA application boosts active zone levels of these proteins. In order to functionally address the role of PKC mediated LLPS for neuron culture active zone function, they establish Liprin- α 2/ α 3 double knockout neuron cultures, where levels of RIM, Munc13, ELKS and the pool of releasable vesicles is found reduced. Moreover, In this background, they use the Liprin- α 3 re-expression mediated rescue to investigate effects of point mutating the Liprin- α 3 phosphorylation site- Indeed, the point mutated form is less effective in rescuing RRP deficits and PMA mediated recruitment of scaffold proteins.

This per se is an interesting study of a highly relevant subject, where causal relations are difficult to establish with previous approaches often entailing full domains of proteins. The study presented here is appealing as the (single) genetic manipulation being used is a site-specific mutation. It is needless to say that consequences different from LLPS might emerge from this point mutation, and that LLPS in the most extreme case might just be an epiphenomenon of some other processes. I am not saying that this is most likely, but I would like them to tone down their statements, particularly "Phosphorylation triggers presynaptic phase separation of Liprin- α 3 TO control active zone structure ". I think it would be more adequate to say "Phosphorylation triggers presynaptic phase separation of Liprin- α 3 AND controls active zone structure ". Quite honestly, the "plasticity paradigm they are using, PMA application, is highly artificial. I would encourage them to clearly speak this out as well.

Given the high quality typical of the Kaeser lab I only do have two additional points:

1. It appears likely that LLPS might also change the accessibility of epitopes. How sure are they that level differences they detect via IF are real? Could they probe via additional antibodies or use on-locus XFP fusions?
2. I found the effects on Synapsin in Figure 4C very interesting. How could they imagine the cross-talk between Synapsin and Liprin-triggered LLPS, particularly concerning SV clustering? Similarly for CaV2.1: how do they explain the increase here?

Reviewer #3:

Remarks to the Author:

This is an interesting manuscript from the Kaeser group exploring a form of PKC triggered synaptic plasticity that is mediated by liprin- α 3. They showed strong evidence that support the following claims:

1. Liprin- α 3 is phosphorylated by PKC on S760.
2. Liprin- α 2, α 3 double knockout show reduction of active zone proteins and reduction of pool of releasable vesicles.
3. Wild type liprin- α 3 can rescue the double KO phenotype while the phosphorylation disabled version of liprin- α 3 cannot.
4. PKC activation by PMA treatment increase active zone proteins and transmitter release.

These are interesting and novel results demonstrating a form of presynaptic plasticity. It would significantly strengthen this part of the paper if the authors can show that this type of presynaptic plasticity can be found in a more physiological setting since the existing experiments relies on PMA treatment.

The authors also showed evidence that the phosphorylation of S760 induce Liprin- α 3 to form phase droplet like puncta in HEK293T cells. They have referred these puncta as phase separated droplets based on FRAP analyses. This part of the manuscript is too preliminary for several reasons. First, only FRAP was used to characterize the potential phase droplets. The phase

separation field uses several additional methods to demonstrate liquid behaviors such as observation of fusion, relaxation and fission. Second, identification of phase separation mechanism should be included. For example, characterization of intrinsically disordered motif(s) or domains will help to establish that these droplets are indeed phase droplets. Third, Liprin-a2 also seem to form puncta but is not phosphorylated by PKC. The relationship between the phase separation between a2 and PKC mediate phase separation of a3 should be characterized. Finally, while the authors present multiple evidences that phosphorylation of S760 regulates potential phase separation of liprin-a3 in 239T cells, these data do not exclude the possibility that this phosphorylation can also affect other functions of liprin-a3, such as the activation of liprin or binding between a3 and other active zone proteins. Therefore, it is premature to draw a strong conclusion between phase separation and the function of liprin-a3.

The manuscript is already very comprehensive and utilizes many cutting-edge techniques. In my opinion, the authors should not make the phase separation a major part of this manuscript unless there is additional evidence to address the issues I raised above. I think it is a nice paper just focusing on the plasticity part if they can find some physiological context. Phase separation should not be in the title and the abstract should not include causal claims such as "We conclude that Liprin-a3 phosphorylation rapidly triggers presynaptic phase separation to modulate active zone structure and function"

Below are my specific comments:

There is not enough evidence to be sure that these puncta are phase droplets. Are there any liquid behavior that can be observed such as fusion and relaxation?

In Fig. 2, does the S760E version of puncta behave similarly in a FRAP assay compared to the wild type puncta?

In ext Fig. 2, liprin-a2 showed constitutive droplet formation in the absence of the S760 phosphorylation. This result suggests liprin-a2 use a different mechanism to form puncta. Since a2 and a3 are the main active zone liprins, how does the a2 behavior affect a3's puncta formation?

In fig.3, liprin-a3 together with RIM1a and Munc13-1 form puncta in the absence of PMA. Do these puncta dissolve when the cells are treated with PKC inhibitor? This experiment will establish that this more physiological relevant condensate use the same phosphorylation mechanism to form and is therefore important.

In Fig. 3i, the authors hint that the phase separation mechanism is responsible for the recruitment of active zone proteins to existing presynaptic terminals in response to PMA treatment. An important issue here is whether the existing active zones exist in a phase separated liquid state. Can FRAP experiments be performed on these active zones in vivo?

It is interesting that the Liprin-a3SG can localize almost normally to the active zone but does not rescue the RIM localization phenotype nor the EPSC charge phenotype (fig. 5u and v). if phase separation of liprin-a3 is required to recruit Munc-13 and required to localize liprin-a3 to active zone, then it should be predicted that liprin-a3SG should not localize to active zone. How can these results be reconciled?

In fig. 6a, how fast is the effect of PMA treatment? From the figure, it seems that the dramatic increase in mEPSC frequency happened within 1 second. This is very short for phase separation to occur and for active zone to grow. How fast is the PMA induced phase droplet formation in non-neuronal cells?

How long is the PMA treatment in fig.7 in order to see an increase in the active zone components? It is also interesting that Liprin-a3SG 's steady state level is not very dramatically different from that of the wt, but it fails to increase upon PMA stimulation. This suggest that phase separation is not required for L-3a to localize to active zone but instead might serve as a specific plasticity mechanism.

The data presented in Fig. 7h and 7i are not consistent with each other. L-a3SG showed increased Munc13-1 level in 7h but no increase in 7i.

Response for Emperador-Melero et al., “Phosphorylation of Liprin- α 3 triggers phase separation and controls presynaptic active zone structure”, NCOMMS-20-40775-T

Summary of new experiments

We thank the reviewers for their insightful comments and for their enthusiasm for our work. In the fully revised version of this manuscript, we have added the new following data and analyses:

1. We show that neither activation nor inhibition of CamKII in HEK293T cells co-transfected with Liprin- α 3 and CamKII α triggers condensate formation (Supplementary Fig. 1b)
2. We added new data on fusion and fission of Liprin- α 3 condensates (Supplementary Figs. 1e, 1f).
3. We quantified relaxation kinetics of newly fused Liprin- α 3 condensates (Figs. 1f, 1g).
4. We performed in vitro phosphorylation assays. They suggest that S764, in contrast to S760, is not phosphorylated by PKC (Supplementary Fig. 2e).
5. We assessed the liquid properties of the Liprin- α 3^{SE} mutant by FRAP, showing that there is dynamic exchange of material (Figs. 2g, 2h).
6. We analyzed the presence of intrinsically disordered regions of all Liprin- α isoforms, indicating that Liprin- α 3 contains highly disordered regions surrounding S760 (Supplementary Figs. 3b, 3c).
7. We generated fluorophore-tagged versions of Liprin- α 1, - α 2 and - α 4 to corroborate that PMA does not alter the distribution of these Liprins (Supplementary Figs. 3d, 3e).
8. We assessed the properties of Liprin- α 2 condensates. Despite forming condensates, it is not exchanged between the condensed phase and the cytosol, indicating that Liprin- α 2 (different from Liprin- α 3) does not behave as a liquid (Supplementary Figs. 3f, 3g).
9. We include new structure-function analyses of Liprin- α 3 condensate formation, which suggest that cooperative interactions between Liprin- α 3 domains are needed (Supplementary Figs. 3h, 3i).
10. We assessed how Liprin- α 2 condensates influence the properties of Liprin- α 3 condensates and how the condensates interact with one another (Supplementary Fig 4).
11. We tested whether PKC inhibitors block PMA-induced phase separation of Liprin- α 3, and found that they do (Supplementary Fig. 5c).
12. We assessed whether PKC inhibitors influence the constitutive formation of condensates containing Liprin- α 3, RIM1 α and Munc13-1 (Supplementary Fig. 5d).
13. We generated a new antibody against Liprin- α 3. The validation for immunolabeling using Liprin- α 3 knockout neurons as controls is included in Supplementary Fig. 6f.
14. We assessed the levels of Liprin- α 3 at the active zone upon PMA addition using this new, independent antibody, and confirm our initial findings (Supplementary Figs. 6g, 6h).

We hope that these **new data**, the **adjustments in wording and interpretation** in the text, and the detailed answers that follow fully address the reviewers’ questions and concerns. Below, we cite the reviewer comments in *black, italic typeface*, add our **responses in blue typeface**, and cite text from the paper in *blue italic typeface with key changes in bold*. We note that the numbering of the cited references is different between this response and the manuscript, and a separate citation list is included in each document.

Responses to Reviewer #1

Accumulating evidence indicate that phase separation is a general mechanism for organizing compartmentalized molecular machineries both in pre- and post-synaptic sides of synapses. In

the presynaptic active zones, RIM and RIM-BP have been shown to undergo phase separation in vitro; Munc13-1 has been observed to form supramolecular nano-assembly in living neurons and non-neuronal cells; ELKS has been detected as cytoplasmic clusters with properties of liquid condensates. In this manuscript, Emperador-Melero et al. report that Liprin- α 3 undergoes phase separation in a PKC phosphorylation-dependent manner. Interestingly, of the four Liprin- α isoforms, only the two that are enriched in presynaptic active zones (α 2 and α 3) undergo phase separation, with α 2 constitutively forming droplets in HEK293T cells and α 3 forming condensates only after PKC-mediated phosphorylation on a single Ser residue. Using CLEM, the authors discovered that RIM and Munc13-1, can form colocalized condensates with Liprin- α 3 beneath the plasma membranes in HEK293T cells, recapitulating the active zone-like structure in synapses. The authors generated Liprin- α 2/ α 3 double knockout mice and performed elegant super-resolution imaging and electrophysiological studies that provide convincing results showing essential roles of Liprin- α 3 and its phosphorylation by PKC in organizing the active zone structure and in controlling neurotransmitter releases in neurons.

We thank the reviewer for this concise summary of our work.

This is a comprehensive and high quality study covering boarded areas including biochemistry, cell biology, and electrophysiology. Most importantly, the authors have been able to demonstrate that formation of protein condensates (using Liprin- α 3 as the example) is critical for the formation and physiological function of active zones in presynaptic boutons. Therefore, the study is very timely and much desired in the field of phase separation in neuronal synapses specifically and in cell biology in general. The study is well designed. Experiments are performed with high quality, and results are clearly described and presented. This reviewer recommends highly of this work for publication in Nature Communication.

We thank the reviewer for this very positive assessment, pointing out the high quality, the broad cell biological approach, the critical nature of our findings, and the timely nature of our work. We are thrilled that the reviewer highly recommends publication in Nature Communications.

I do have a few minor issues for the authors to consider:

1. Figure 1a: the authors used inhibitors and activators of PKA, PKC and CaMKII, and only PKC activation led to Liprin- α 3 droplets formation. But the expression of CaMKII in HEK293T cells is low, so it's perhaps not surprising that the cells do not respond to CaMKII activator or inhibitor. A possible experiment to test potential impact of CaMKII phosphorylation is to co-transfect Liprin- α 3 and CaMKII in HEK293T cells.

We agree with the reviewer's concern and performed the suggested experiment. We co-transfected HEK293T cells with CamKII α and Liprin- α 3 and added either caffeine (a CamKII activator) or KN-93 (a CamKII inhibitor). These manipulations did not result in the formation of Liprin- α 3 condensates (Supplementary Fig. 1b), supporting the conclusion that CamKII is unlikely to phosphorylate Liprin- α 3 to trigger its phase separation.

As suggested by the Extended data figure 2b&2c, S764 phosphorylation also contributes to Liprin- α 3 phase separation, implying other kinase(s) may participate in the regulation of Liprin- α 3 phase separation.

We thank the reviewer for this comment and realize that we did not describe the data on S764 appropriately. The S764A mutation blocks PMA-induced phase separation of Liprin- α 3, indicating that this position is relevant for this process. Three possibilities may explain this effect:

- S764 may be directly phosphorylated by PKC. We performed a new experiment that provides strong evidence against this possibility (Supplementary Fig. 2e): purified Liprin- α 3 incubated with recombinant PKC does not show an increase in phosphorylation at this site (detected with phospho-S764 antibodies). Furthermore, phospho-S764 antibodies also do not detect a signal increase in HEK293T cells transfected with Liprin- α 3 and incubated with PMA (Supplementary Fig. 2d). In contrast, phospho-S760 antibodies detect a strong signal increase in both assays (Fig. 2c and Supplementary Fig. 2e). This supports quite strongly that S764 is not phosphorylated by PKC, and matches well with the point that S760 is surrounded by a PKC consensus sequence while S764 is not ¹.
- Other kinases may phosphorylate S764 to trigger phase separation. We induce phase separation by PMA addition, which is a PKC activator. We now show in new experiments that when PMA is added in the presence of a PKC blocker, phase separation of Liprin- α 3 is prevented (Supplementary Fig. 5c). Hence, it is unlikely that kinases other than PKC are responsible for Liprin- α 3 phase separation upon PMA addition.
- Finally, S764 may participate indirectly in phase separation. Given that our experiments largely exclude the two possibilities above, we think that this is the most likely explanation. One possibility is that S764 is part of the PKC recognition sequence and mutating it could hamper phosphorylation of S760. In fact, serine is the most common residue in PKC targets at the + 4 position ¹. Another possibility is that S764 participates in the intramolecular changes that are necessary for phase separation without participating in phosphorylation, for example through contributing to potential intramolecular interactions of Liprin- α ².

We hope that this clarifies this point. We have now adjusted the text (lines 144-156 and 209-210) to better express these possibilities. Even if S764 PKC phosphorylation occurs as well despite our failure to detect it and despite the notion that S764 is not surrounded by a PKC consensus, we note that S760 is both sufficient and necessary to drive phase separation in HEK293T cells, it mediates PMA-induced changes in synaptic structure and transmission, and S760 phosphorylation can be readily detected while S764 phosphorylation cannot be identified. Hence, the overall conclusion that S760 phosphorylation by PKC mediates phase separation and controls active zone structure is not affected by the remaining uncertainty about S764.

2. Extended data figure 2f: both Liprin- α 2 and Liprin- α 3 (+PMA) forms “ring-like” structures instead of “droplet-like” structure shown in other figures. What are these “ring-like” structures?

We thank the reviewer for raising this point. Throughout, we assess condensate formation in transfected HEK293T cells with fluorophore-tagged proteins, and we observe condensates as “droplets”. In the previous version of the manuscript, the only exception was the assessment of Liprin- α 1, - α 2 and - α 4 because we had not generated tagged proteins. We instead used antibody labeling, and we observed that antibodies did not penetrate the phase (after fixation), leading to ring-like structures. We note that this is the case for antibody labeling of fluorescently tagged Liprin- α 3, as droplets are detected when imaging the fluorophore, but rings are seen when imaging antibody-labeled droplets (Supplementary Figs. 1c, 1d).

To address this point and to make the manuscript clearer, we have now generated fluorophore-tagged versions of each Liprin- α protein and assessed condensate formation in the presence and absence of PMA (Supplementary Figs. 3d, 3e). We obtained the same results that we observed with antibody labeling, which is that Liprin- α 1 and - α 4 rarely form condensates, - α 2 forms constitutive condensates (which do not have liquid properties), and - α 3 forms liquid condensates only upon PMA addition. In the revised manuscript, we have replaced the previous antibody labeling data (“rings”) with the data of the tagged Liprin- α proteins (Supplementary Figs. 3d, 3e).

3. *Extended data figure 3a: Munc13-1 directly interacts with RIM but is not known to bind to Liprin-α3. Curiously, Munc13-1 formed co-localized patches with Liprin-α3 but not with the RIM droplets. Additionally, co-transfection of Munc13-1 with Liprin-α3 resulted in very striking morphology changes of the Liprin-α3 condensates. Maybe the authors can comment on this observation that is rather counterintuitive.*

We thank the reviewer for bringing this up and agree with the assessment that this is unexpected because no direct interaction between Munc13-1 and Liprin-α3 is known. We commented on it in the discussion (lines 421-429) to state that direct interactions between these proteins are not known, but that the data fit quite well with the observation that Liprin-α deletion in flies leads to loss of a specific Munc13 isoform from at the neuromuscular junction³, and our observation that deletion of the main presynaptic Liprin-α isoforms causes partial loss of Munc13-1 (Figs. 4b, 4c, 5e-5h). We think that there might either be unknown direct interactions between these proteins, or that there are adapters between them that are present both at synapses and in HEK293T cells.

4. Figure 6: is there any reason why the KO^{L23} control is not included in the figure.

The purpose of this experiment (current Fig. 7) was to determine whether S760 phosphorylation of Liprin-α3 is important for active zone structure and function. Electrophysiological recordings demand multiple coverslips per group and require to perform all recordings within a narrow, ~2-day time window for a given culture. Furthermore, the PMA-experiments only allow for a 20 min time window of recording per coverslip (see methods, lines 754-759), and hence require a large number of coverslips per experimental condition. Furthermore, we believe strongly that the most rigorous way to perform these group comparisons is to collect all groups from each culture with approximately even numbers of observations per group and across cultures.

Because of these points, comparing KO^{L23}, KO^{L23} + Liprin-α3, and KO^{L23} + Liprin-α3^{SG}, with or without PMA addition (six groups in total), was not feasible. We instead directly compared the essential conditions (KO^{L23} + Liprin-α3 vs KO^{L23} + Liprin-α3^{SG}, with and without PMA, four groups). We note that this is different in STED experiments (current Fig. 8), where one coverslip per condition and culture is sufficient and the time window of analysis is not narrow, making a 6-group experiment feasible. This is the reason why the KO^{L23} group was included for STED, but not for electrophysiology experiments. Notably, we provide a complete comparison of rescue (KO^{L23} + Liprin-α3) to essential controls (control^{L23} and KO^{L23}, Figs. 6s-6w) and additional comparisons (Supplementary Fig. 9). We hope that the reviewer understands that it is not always possible to record all control conditions in each experiment at the same time because of experimental constraints (recording time, amount of available material).

We believe that the overall conclusion that the phosphorylation site is important for the function of Liprin in active zone structure and function is accurate because directly comparing KO^{L23}, KO^{L23} + Liprin-α3, and KO^{L23} + Liprin-α3^{SG} at basal conditions shows a striking lack of rescue in the KO^{L23} + Liprin-α3^{SG} condition (Figs. 6s-6w). We hope that the reviewer agrees that the overall conclusion is valid with the current experimental design.

Responses to Reviewer #2

The last years have seen the emergence of liquid-liquid phase separation (LLPS) as a new concept to organize membrane-less compartments. Both assembly but also dynamic regulation

of the presynaptic active zone scaffold might well be organized by LLPS. Indeed, a previous study suggested that RIM, RIM-BP as well as the VGCC C-term might engage into reversible liquid-liquid phase separation. However, direct genetic evidence to document the regulatory role of LLPS is missing and not necessarily easy to establish, so that the question whether LLPS indeed is a regulatory principle still pertains.

In this study, the Kaeser lab investigates active zone scaffold proteins in the LLPS context in HEK cells and cultivated hippocampal neurons. In HEK cells, PMA application (PLC-induced generation of diacylglycerol and activates PKC) provoked the reversible formation of Liprin- α 3 spherical condensates indicative of LLPS. FRAP analysis demonstrated liquid like features of the Liprin- α 3 condensates. Again, using HEK cells, they find that PKC phosphorylation of serine-760 (S760) of Liprin- α 3 triggers Liprin α 3 LLPS “in vivo”. S760A 136 and S764A Liprin- α 3 mutations interfered with PMA-induced droplet formation in HEK cells. The detect in cellulo phosphorylation via anti-phospho-S760/764 antibodies.

They then turn to cultured hippocampal neurons and assess active zone levels of endogenous Liprin- α 3, RIM and Munc13-1 at synapses using STED microscopy. They show that PMA application boosts active zone levels of these proteins. In order to functionally address the role of PKC mediated LLPS for neuron culture active zone function, they establish Liprin- α 2/ α 3 double knockout neuron cultures, where levels of RIM, Munc13, ELKS and the pool of releasable vesicles is found reduced. Moreover, In this background, they use the Liprin- α 3 re-expression mediated rescue to investigate effects of point mutating the Liprin- α 3 phosphorylation site- Indeed, the point mutated form is less effective in rescuing RRP deficits and PMA mediated recruitment of scaffold proteins.

We thank the reviewer for taking the time to review our paper and to accurately summarize the key findings.

This per se is an interesting study of a highly relevant subject, where causal relations are difficult to establish with previous approaches often entailing full domains of proteins. The study presented here is appealing as the (single) genetic manipulation being used is a site-specific mutation.

We are thankful that the reviewer finds our work interesting and highly relevant. We further naturally agree with the reviewer on the appealing nature of our approach in which a single point mutation has a strong cell-biological effect.

It is needless to say that consequences different from LLPS might emerge from this point mutation, and that LLPS in the most extreme case might just be an epiphenomenon of some other processes. I am not saying that this is most likely, but I would like them to tone down their statements, particularly “Phosphorylation triggers presynaptic phase separation of Liprin- α 3 TO control active 4 zone structure “. I think it would be more adequate to say “Phosphorylation triggers presynaptic phase separation of Liprin- α 3 AND controls active zone structure “. Quite honestly, the “plasticity paradigm they are using, PMA application, is highly artificial. I would encourage them to clearly speak this out as well.

*We fully agree with the reviewer’s assessment. Despite the multifaceted nature of the approach and the quite subtle molecular manipulations (a single S-to-G point mutation), causality cannot be established with 100% certainty. As suggested by the reviewer, we have adjusted to title from “to” to “and”, the new title now is “**Phosphorylation of Liprin- α 3 triggers phase separation and controls presynaptic active zone structure**”. We have also modified the abstract and wording throughout the manuscript to tone down claims of causality. Instead, we present the possibility that Liprin- α 3 phase separation regulates active zone structure as a*

working model.

We also agree that, while widely used to study synapses, it remains uncertain how exactly PMA and other DAG analogues relate to physiological pathways. This is a common limitation in many previous studies (for examples see ⁴⁻¹³). We have included a statement to clarify this important point (lines 339-340): "...***it remains uncertain how potentiation induced by phorbol esters relates to physiological synaptic plasticity.***"

Given the high quality typical of the Kaeser lab I only do have two additional points:

We thank the reviewer for the generous compliment and the appreciation of the quality of our work.

1. It appears likely that LLPS might also change the accessibility of epitopes. How sure are they that level differences they detect via IF are real? Could they probe via additional antibodies or use on-locus XFP fusions?

We thank the reviewer for pointing this out. At STED resolution, the levels of Liprin- α 3 at the active zone increase upon PMA in wild type synapses using an antibody against endogenous Liprin- α 3 (Figs. 3i, 3j). Similarly, in KO^{L23} synapses rescued with HA-tagged Liprin- α 3, antibody signals increase with an antibody against the HA epitope (Figs. 6v, 6w).

We now have added another experiment in wild type neurons with a newly generated, independent Liprin- α 3 antibody as suggested by the reviewer. The antibody generation is described in the methods (lines 580-589), and the antibody is validated for immunostainings in control and Liprin- α 3 knockout neurons in Supplementary Fig. 6f. We note that while the antibody in Figs. 3i, 3j is directed against a short peptide ¹⁴, the new antibody is a polyclonal antibody with a much larger antigen (amino acids 188-576, see methods), which would likely improve antigen accessibility strongly if this was a limitation in the initial experiments. Using this new antibody, we detected an increase in Liprin- α 3 (Supplementary Figs. 6g, 6h) similar to the increase observed in the initial experiment and in the rescue experiment (Figs. 3i, 3j, 6v, 6w). Hence, we have three independent experiments performed with three independent antibodies, directed against multiple areas of the Liprin- α 3 protein, all reporting qualitatively and quantitatively similar changes. We feel strongly that this establishes that the change is real and alleviates concerns regarding epitope accessibility.

An increase of similar magnitude was observed for Munc13-1 and RIM. Unfortunately, additional antibodies for the other synaptic proteins suitable for STED superresolution are not available. We note, however, that the antibodies we use have been validated and used for immunostaining across methods and preparations and in many experiments ¹⁵⁻²³. Furthermore, the experiments with fluorophore-tagged proteins in HEK293T cells fit the experiments with antibodies at synapses qualitatively quite well. Whether the exact quantification is correct remains naturally somewhat uncertain, as it does for all studies that assess proteins with antibodies, due to the nature of the method. We hope that the reviewer agrees that the fundamental effect on Liprin- α 3 is well established with localizing both endogenous and transduced Liprin- α 3 and finding remarkably similar effects with three independent antibodies.

(C3) 2. I found the effects on Synapsin in Figure 4C very interesting. How could they imagine the cross-talk between Synapsin and Liprin-triggered LLPS, particularly concerning SV clustering? Similarly for CaV2.1: how do they explain the increase here?

We thank the reviewer for pointing out these observations. We now discuss the change in Cav2.1 on lines 413-421 and those of Synapsin on lines 431-441. Briefly, we propose that several protein complexes (or liquid condensates) may exist at the active zone. We speculate that, while a complex containing RIM, RIM-BP and Cav2 functions in tethering channels, a different one formed by Liprin- α , RIM and Munc-13 regulates docking and may mark release sites. We think that this is in line with independent studies that suggest that Munc13s and RIMs mark release sites, but may not be strictly colocalized with Cav2s^{3,24-26}. Similar principles may also apply to vesicle clustering, for which Synapsin plays a major role. We think that Liprin- α proteins may be present in multiple phases, and perhaps may even bridge phases or attach them to one another. Removing one phase may further induce compensatory changes in the other, as they may be in equilibrium with one another or compete for space. We note that these models are speculative and will require future studies. We hope that our work inspires others to help working out these models.

Responses to Reviewer #3

This is an interesting manuscript from the Kaeser group exploring a form of PKC triggered synaptic plasticity that is mediated by liprin-a3.

They showed strong evidence that support the following claims:

- 1. Liprin-a3 is phosphorylated by PKC on S760.*
- 2. Liprin-a2, a3 double knockout show reduction of active zone proteins and reduction of pool of releasable vesicles.*
- 3. Wild type liprin-a3 can rescue the double KO phenotype while the phosphorylation disabled version of liprin-a3 cannot.*
- 4. PKC activation by PMA treatment increase active zone proteins and transmitter release. These are interesting and novel results demonstrating a form of presynaptic plasticity.*

We thank the reviewer for the overall positive assessment and for highlighting the strength of our evidence for multiple aspects of Liprin- α and presynaptic function.

It would significantly strengthen this part of the paper if the authors can show that this type of presynaptic plasticity can be found in a more physiological setting since the existing experiments relies on PMA treatment.

We thank the reviewer for raising this point. We think that this reflects the general limitation in the field. Namely, despite decades of wide use (for a few examples, see⁴⁻¹³), it remains unknown how phorbol ester-induced potentiation relates to physiological plasticity. Unfortunately, it is not currently possible to determine how to induce this pathway with a physiological paradigm. For this reason, we believe this is a valid limitation for the part of the manuscript that uses PMA in neurons (Figs. 3i, 3j, 7, 8), as it has been a valid limitation for many other influential studies before. We have now clearly state this limitation (lines 338-340): ***“...it remains uncertain how potentiation induced by phorbol esters relates to physiological synaptic plasticity.”***

We emphasize that only a subset of our experiments in synapses relies on PMA, and that our overall conclusion that phosphorylation is important for the synaptic function of Liprin- α 3 is supported by several experiments on synapse structure and function independent of PMA treatment. We establish that knockout of Liprin-a2/3 induces structural (Figs. 4 & 5) and functional (Figs. 6a-6r) synaptic defects, and that phenotypes are rescued by wild-type Liprin- α 3 (Supplementary Figs. 9a-9h), but not by Liprin- α 3 carrying a point mutation (S760G) that blocks

PKC phosphorylation (Figs. 6t-6w).

We hope that the reviewer understands that this problem can currently not be solved, but that many important findings on synaptic function are based on similar approaches⁴⁻¹³. We feel that the method can be used when limitations are acknowledged and alternative experiments establish that phenotypes do not solely depend on PMA treatment, as is the case in our study.

The authors also showed evidence that the phosphorylation of S760 induce Liprin-α3 to form phase droplet like puncta in HEK293T cells. They have referred these puncta as phase separated droplets based on FRAP analyses. This part of the manuscript is too preliminary for several reasons. First, only FRAP was used to characterize the potential phase droplets. The phase separation field uses several additional methods to demonstrate liquid behaviors such as observation of fusion, relaxation and fission.

We thank the reviewer for raising this important point. As suggested by the reviewer, we performed additional experiments. We now show that fusion and fission of Liprin-α3 droplets occur (Supplementary Figs. 1e, 1f), and we quantitatively assess relaxation dynamics of Liprin-α3 condensates after fusion (Figs. 1f, 1g). These new data are described on lines 107-122. Overall, our results match very well with those of other liquid condensates^{27,28}. Along with the lack of membrane enclosure observed by CLEM and the dynamic exchange of material observed by FRAP, we believe that the data are now very strong to establish liquid condensate formation within cells.

We thank the reviewer for pointing out that we needed a better assessment of the fundamental phenomenon of Liprin-α3 phase separation, and we think that this suggestion has greatly strengthened the manuscript.

Second, identification of phase separation mechanism should be included. For example, characterization of intrinsically disordered motif(s) or domains will help to establish that these droplets are indeed phase droplets.

Again, we thank the reviewer for pointing this out. We now include an analysis of Liprin-α3 sequences identifying intrinsically disordered regions (IDRs) in the area of the S760 PKC phosphorylation site (Supplementary Figs. 3a, 3b). This region is well conserved between human, mouse and rat Liprin-α3. Notably, similarly disordered regions are also present in the other Liprin-α proteins (Supplementary Fig. 3c), which do not form liquid condensates in our experimental paradigm. These analyses are described on lines 167-176, and suggests that the IDR alone is not sufficient to mediate phase separation.

We have performed new experiments to further address this working model. We have expressed Liprin-α3 fragments covering full-length Liprin-α3, and defined boundaries by their structure, homology, and known interactions²⁹⁻³². We found that no individual fragment undergoes phase separation, including a protein that encompasses amino acids 577-790 and contains the IDR, either at baseline or after PMA application (Supplementary Figs. 3h, 3i, described on lines 179-183). This indicates that larger Liprin-α3 sequence elements, spanning multiple domains are necessary for this regulated form of phase separation. This observation is well in line with the working model that intramolecular interactions control Liprin-α function².

We thank the reviewer for bringing up this point, as we think it strengthens the argument that phosphorylation induces changes in the overall state of Liprin-α3 such that it can undergo phase separation.

Third, Liprin-a2 also seem to form puncta but is not phosphorylated by PKC. The relationship between the phase separation between a2 and PKC mediate phase separation of a3 should be characterized.

This is a great suggestion and we performed new experiments to characterize the Liprin- α 2 condensates and their interaction with Liprin- α 3 condensates. Surprisingly, the Liprin- α 2 condensates do not have liquid properties, and we are very thankful that the reviewer pointed out the need to characterize the Liprin- α 2 puncta better.

We generated a fluorophore-tagged version of Liprin- α 2 to assess its properties via FRAP (Supplementary Figs. 3f, 3g). Despite formation of condensate-like, round structures, Liprin- α 2 is not dynamically exchanged with the cytosol as assessed by FRAP. This behavior is very different from that of Liprin- α 3 condensates. Hence, Liprin- α 2 condensates do not follow liquid dynamics. These data are described in the main text of the manuscript on lines 177-178. More broadly, the notion that we observe different biophysical properties supports the model of diversification of function between Liprin- α proteins that is also seen in genetic experiments (discussed on lines 391-407), and further indicates that our experimental approach is suitable to detect such differences.

To assess the relationship between Liprin- α 2 and - α 3 condensates, we co-transfected HEK293T cells with these proteins. Before PMA addition, Liprin- α 2 and - α 3 are co-recruited into condensates, with enhanced presence of Liprin- α 2 in the core of the condensates and Liprin- α 3 in their periphery (Supplementary Figs. 4a, 4b). While Liprin- α 3 is dynamically exchanged with the cytosol (as revealed by FRAP analysis), Liprin- α 2 is not (Supplementary Figs. 4c, 4d). When treated with PMA, the number of condensates containing both proteins increased ~3.5 fold (Supplementary Figs. 4e, 4f). These data, discussed on lines 185-189 of the manuscript, reveal that there is interplay between various Liprin- α proteins, as has been suggested before in biochemical and genetic experiments^{14,16,33,34}.

Finally, while the authors present multiple evidences that phosphorylation of S760 regulates potential phase separation of liprin-a3 in 293T cells, these data do not exclude the possibility that this phosphorylation can also affect other functions of liprin-a3, such as the activation of liprin or binding between a3 and other active zone proteins. Therefore, it is premature to draw a strong conclusion between phase separation and the function of liprin-a3. The manuscript is already very comprehensive and utilizes many cutting-edge techniques. In my opinion, the authors should not make the phase separation a major part of this manuscript unless there is additional evidence to address the issues I raised above. I think it is a nice paper just focusing on the plasticity part if they can find some physiological context. Phase separation should not be in the title and the abstract should not include causal claims such as “We conclude that Liprin- α 3 phosphorylation rapidly triggers presynaptic phase separation to modulate active zone structure and function”.

We thank the reviewer for pushing us to perform a better characterization of Liprin- α phase separation (including fusion and fission, relaxation, IDR analyses, structure-function analyses of Liprin- α 3 phase separation, better assessment of Liprin- α 1, Liprin- α 2, and Liprin- α 4, and interactions between condensates). We hope that the reviewer agrees that the inclusion of all the suggested experiments strengthen the manuscript significantly and validate the conclusion that Liprin- α 3 undergoes liquid-liquid phase separation.

We fully agree with the reviewer that we cannot be absolutely certain about the causative

relationship between phase separation and structural roles at synapses. To express this uncertainty, we have adjusted the title to remove the claim of causality (current title: "Phosphorylation of Liprin- α 3 triggers phase separation **AND** controls presynaptic active zone structure"), and have modified the abstract and main text to avoid direct causality claims. Instead, we discuss the relationship between phase separation and active zone structure and present our findings as a working model, and we note specifically that S760 could also function in a phase separation-independent way (lines 381-382: "... these mechanisms of S760 could further function **in phase-separation-independent ways**"). Ultimately, there is no doubt that this area of biology, both at synapses and in other cellular contexts, requires future research to better develop the model that phase separation controls cellular functions.

We hope that the reviewer agrees that with the new data on phase separation added and the causality claim removed, the biological phenomena we find are presented appropriately.

Below are my specific comments:

There is not enough evidence to be sure that these puncta are phase droplets. Are there any liquid behavior that can be observed such as fusion and relaxation?

As outlined above, we have added new data to establish that fusion, fission and relaxation events occur (Figs. 1f, 1g and Supplementary Figs. 1e, 1f). We believe that these new data significantly strengthen the conclusion that Liprin- α 3 droplets are liquid condensates.

In Fig. 2, does the S760E version of puncta behave similarly in a FRAP assay compared to the wild type puncta?

We have carried out a new experiment to address the question whether the phosphomimetic point mutation generates droplets with liquid properties. The fluorescence of Liprin- α 3 S760E droplets recovers with properties similar to those of Liprin- α 3 (Figs. 2g, 2h), and the new data are described on lines 162-164.

In ext Fig. 2, liprin- α 2 showed constitutive droplet formation in the absence of the S760 phosphorylation. This result suggests liprin- α 2 use a different mechanism to form puncta. Since α 2 and α 3 are the main active zone liprins, how does the α 2 behavior affect α 3's puncta formation?

We thank the reviewer for pointing this out. Indeed, and as outlined above, Liprin- α 2 condensate properties are very different from Liprin- α 3 droplets. mVenus-tagged Liprin- α 2, despite forming condensates, is not exchanged with the cytosol (no recovery in FRAP analyses, Supplementary Figs. 3f, 3g), indicating that the condensates are not liquids.

When co-transfected, Liprin- α 2 and - α 3 are co-recruited to the same condensates, where - α 2 primarily occupies the core and - α 3 the periphery under basal conditions, and Liprin- α 3 is dynamically exchanged with the cytosol, but Liprin- α 2 is not (Supplemental Figs. 4a-4d). Furthermore, addition of PMA increases the number of condensates containing Liprin- α 2 and - α 3 (Supplementary Figs. 4e, 4f). This suggests that there is interplay between Liprin- α proteins.

In fig.3, liprin- α 3 together with RIM1a and Munc13-1 form puncta in the absence of PMA. Do these puncta dissolve when the cells are treated with PKC inhibitor? This experiment will establish that this more physiological relevant condensate use the same phosphorylation mechanism to form and is therefore important.

We thank the reviewer for proposing this experiment and performed. Neither chronic nor acute addition of the PKC inhibitor Bis-1 to HEK293T cells expressing Liprin- α 3, Munc13-1 and RIM1 α blocked the formation of condensates in the absence of PMA (Supplementary Fig. 5d). This indicates that PKC-independent mechanisms contribute to the formation of these condensates. It aligns well with the observations that (1) active zone protein complexes containing Liprin- α 3, RIM and Munc13 exist under basal conditions at synapses without PKC activation (Figs. 3i, 3j), (2) Liprin- α 2/3 knockout synapses present defects in active zone structure at baseline conditions (Figs. 5a-5p), and (3) constitutive PKC knockout mice survive³⁵ (which is not compatible with a major role in baseline active zone assembly, as mouse knockout for active zone proteins is typically lethal).

These findings highlight that, while there is baseline assembly of active zones that is independent of the PKC/Liprin- α 3 axis, this axis can actively modulate this process. In the revised manuscript, this is discussed in detail on lines 443-468. We hope that the reviewer agrees that it would be surprising if PKC inhibitors block all condensate formation. Through our and previous work it is well established that PKC, Liprin- α 2 and Liprin- α 3 are not required for active zone assembly. In fact, it is unlikely that there is a single master organizer in the form of a single protein, protein domain, or phosphorylation site for such assembly³⁶. Nevertheless, our data clearly establish that Liprin- α , its phosphorylation, and presumably its phase separation enhance active zone assembly and neurotransmitter release both at baseline and under activation of PLC/PKC signaling.

In Fig. 3i, the authors hint that the phase separation mechanism is responsible for the recruitment of active zone proteins to existing presynaptic terminals in response to PMA treatment. An important issue here is whether the existing active zones exist in a phase separated liquid state. Can FRAP experiments be performed on these active zones in vivo?

The reviewer raises an excellent point that relates to a key limitation in the field. Indeed, addressing liquid properties of active zone condensates in synapses is very important.

Unfortunately, imaging technology accurate enough to answer this question is not available. A typical central nervous synapse is a small structure (approximately 1 μ m in diameter) and the active zone is a very small compartment within it (less than 50 nm thick). There is currently no technology that allows selective FRAP of such a small area within a synapse. Published FRAP studies photo-bleached entire synapses due to this limitation. While these studies reported important dynamics of replenishment of the synaptic pool of molecules from distant areas, they cannot selectively measure exchange of molecules between active zones and the presynaptic cytosol, which is the key parameter for answering whether active zones are liquid condensates within synapses.

Hence, addressing the liquid properties of active zone complexes within a synapse is currently not doable. Instead, available methods bleach both the phase (if it exists) and the recovery pool of proteins, and FRAP reflects refilling of the recovery pool of molecules from distant compartments rather than exchange between that pool and the liquid phase. We state this general limitation in the manuscript on lines 352-354 (“***The lack of technology to photobleach active zones of synapses with the necessary resolution of tens of nanometers prevents performing FRAP experiments to directly answer ...***”). We hope that future technology development will find approaches to circumvent this limitation.

Finally, we would like to note that this limitation expands to other synaptic phases and liquid phases in general. A fundamental limitation in the field is that phase separation can be well

studied with purified proteins and in heterologous cells, but is difficult to assess in small subcellular compartments. We feel that having control over phase separation properties with a single, subtle point mutation in a large protein like Liprin- α is a major step forward. However, some limitations remain, and we hope that technology development will allow us to address this question in the future.

It is interesting that the Liprin-a3SG can localize almost normally to the active zone but does not rescue the RIM localization phenotype nor the EPSC charge phenotype (fig. 5u and v). If phase separation of liprin-a3 is required to recruit Munc-13 and required to localize liprin-a3 to active zone, then it should be predicted that liprin-a3SG should not localize to active zone. How can these results be reconciled?

We thank the reviewer for bringing this up and we apologize that we did not make this clearer in the previous version of the manuscript. The observation of the reviewer that Liprin- α ^{SG} localizes to active zones is correct, and we think that this finding is not in conflict with a lack of recruitment of Munc13 by Liprin- α ^{SG}. Instead, it highlights that active zone recruitment of Liprin- α 3 and phase separation of Liprin- α 3 are distinct processes. Our working model is that Liprin- α 3 phase separation, not Liprin- α 3 targeting to synapses, mediates recruitment of downstream proteins.

The Liprin- α ^{SG} mutation, which abolishes phosphorylation and phase separation, is recruited to the active zone area (although with a mild impairment, Supplementary Figs. 9j, 9k). This mutant retains binding to proteins present at the active zone, for example ELKS and LAR-RPTPs, which may drive Liprin- α 3 active zone targeting. Once at the active zone, phase separation of Liprin- α 3 promotes the recruitment of additional active zone components, for example RIM and Munc13, to strengthen synapses. Hence, a key of our model is that recruitment of Liprin- α 3 to the active zone and enhancement of active zone structure and function by Liprin- α 3 are different processes. These points are discussed in detail in lines 443-468.

(S7) In fig. 6a, how fast is the effect of PMA treatment? From the figure, it seems that the dramatic increase in mEPSC frequency happened within 1 second. This is very short for phase separation to occur and for active zone to grow. How fast is the PMA induced phase droplet formation in non-neuronal cells?

We realize that we did not clearly convey this information in the initial submission and thank the reviewer for pointing this out. The “-PMA” and “+PMA” conditions were recorded from independent coverslips. Neurons were incubated either with PMA (“+PMA”) or with the equivalent amount of DMSO (“-PMA”) for 20 minutes before recordings were performed.

Effects of phorbol esters are typically induced within tens of seconds to minutes, with the full extent reached often within ~5 min (for an example, see Fig. 1 of ¹¹). This time course fits with droplet formation in HEK293T cells, where the first droplets are typically seen within ~1 min, and the full effect plateaus after ~ 15 min (Supplementary Fig. 1a, Supplementary Movie 1). The time line of the electrophysiological experiment is now clearly stated in the manuscript (lines 340-341) and described in detail in the methods section (lines 754-759).

How long is the PMA treatment in fig.7 in order to see an increase in the active zone components? It is also interesting that Liprin-a3SG 's steady state level is not very dramatically different from that of the wt, but it fails to increase upon PMA stimulation. This suggest that phase separation is not required for L-3a to localize to active zone but instead might serve as a specific plasticity mechanism.

We completely agree with the reviewer that phase separation likely serves as a plasticity mechanism, and have mentioned this in the answers to several points above. We propose, as shown in Fig. 8j, that phase separation of Liprin- α 3 is a mechanism that exists on top of constitutive assembly pathways that operate independently of Liprin- α 3 phase separation. In the revised version of the manuscript, this is discussed on lines 451-457. The time course of PMA treatment in Fig. 8 (previous Fig. 7) is 20 minutes before fixation followed by staining, matching the electrophysiological studies as outlined in the point above.

The data presented in Fig. 7h and 7i are not consistent with each other. L- α 3SG showed increased Munc13-1 level in 7h but no increase in 7i.

We apologize that we did not make this clear in the text. The data in these panels (current Fig. 8h, 8i) are fully consistent with one another.

In 8h, raw fluorescence (in arbitrary units, AUs) are shown. Here, KO^{L23} and KO^{L23}+L- α 3^{SG} each show a small increase, while KO^{L23}+L- α 3^{SG} shows a larger increase. In 8i, the data for each group are shown normalized to the corresponding genetic condition without PMA. Again, KO^{L23} and KO^{L23}+L- α 3^{SG} show a small increase (~15%), while KO^{L23}+L- α 3^{SG} shows a 32% increase. We now describe this with more precision than in the previous manuscript on lines 358-361 and hope that this clarifies this point.

References for response to reviewers

1. Kreegipuu, A., Blom, N., Brunak, S. & Järvi, J. Statistical analysis of protein kinase specificity determinants. *FEBS Lett.* **430**, 45–50 (1998).
2. Chia, P. H., Patel, M. R., Wagner, O. I., Klopfenstein, D. R. & Shen, K. Intramolecular regulation of presynaptic scaffold protein SYD-2/liprin- α . *Mol. Cell. Neurosci.* **56**, 76–84 (2013).
3. Böhme, M. A. *et al.* Active zone scaffolds differentially accumulate Unc13 isoforms to tune Ca²⁺ channel-vesicle coupling. *Nat. Neurosci.* **19**, 1311–1320 (2016).
4. Basu, J., Betz, A., Brose, N. & Rosenmund, C. Munc13-1 C1 domain activation lowers the energy barrier for synaptic vesicle fusion. *J. Neurosci.* **27**, 1200–1210 (2007).
5. Malenka, R. C., Madison, D. V & Nicoll, R. A. Potentiation of synaptic transmission in the hippocampus by phorbol esters. *Nature* **321**, 175–7 (1986).
6. Lackner, M. R., Nurrish, S. J. & Kaplan, J. M. Facilitation of synaptic transmission by EGL-30 Gq α and EGL-8 PLC β : DAG binding to UNC-13 is required to stimulate acetylcholine release. *Neuron* **24**, 335–346 (1999).
7. Wierda, K. D., Toonen, R. F., de Wit, H., Brussaard, A. B. & Verhage, M. Interdependence of PKC-dependent and PKC-independent pathways for presynaptic plasticity. *Neuron* **54**, 275–290 (2007).
8. Betz, A. *et al.* Munc13-1 is a presynaptic phorbol ester receptor that enhances neurotransmitter release. *Neuron* **21**, 123–136 (1998).
9. Chang, C. Y., Jiang, X., Moulder, K. L. & Mennerick, S. Rapid activation of dormant presynaptic terminals by phorbol esters. *J. Neurosci.* **30**, 10048–10060 (2010).
10. Gekel, I. & Neher, E. Application of an Epac activator enhances neurotransmitter release at excitatory central synapses. *J. Neurosci.* **28**, 7991–8002 (2008).
11. de Jong, A. P. H. *et al.* Phosphorylation of synaptotagmin-1 controls a post-priming step in PKC-dependent presynaptic plasticity. *Proc. Natl. Acad. Sci. U. S. A.* **113**, 5095–5100 (2016).

12. Rhee, J. S. *et al.* Beta phorbol ester- and diacylglycerol-induced augmentation of transmitter release is mediated by Munc13s and not by PKCs. *Cell* **108**, 121–133 (2002).
13. Taschenberger, H., Woehler, A. & Neher, E. Superpriming of synaptic vesicles as a common basis for intersynapse variability and modulation of synaptic strength. *Proc. Natl. Acad. Sci. U. S. A.* **113**, E4548-57 (2016).
14. Zürner, M., Mittelstaedt, T., tom Dieck, S., Becker, A. & Schoch, S. Analyses of the spatiotemporal expression and subcellular localization of liprin- α proteins. *J. Comp. Neurol.* **519**, 3019–3039 (2011).
15. de Jong, A. P. H. *et al.* RIM C2B Domains Target Presynaptic Active Zone Functions to PIP2-Containing Membranes. *Neuron* **98**, 335-349.e7 (2018).
16. Wong, M. Y. *et al.* Liprin- α 3 controls vesicle docking and exocytosis at the active zone of hippocampal synapses. *Proc. Natl. Acad. Sci.* **115**, 2234–2239 (2018).
17. Held, R. G. *et al.* Synapse and Active Zone Assembly in the Absence of Presynaptic Ca²⁺ Channels and Ca²⁺ Entry. *Neuron* **107**, 667-683.e9 (2020).
18. Wang, S. S. H. *et al.* Fusion Competent Synaptic Vesicles Persist upon Active Zone Disruption and Loss of Vesicle Docking. *Neuron* **91**, 777–791 (2016).
19. Banerjee, A. *et al.* Molecular and functional architecture of striatal dopamine release sites. *bioRxiv* 2020.11.25.398255 (2020). doi:10.1101/2020.11.25.398255
20. Liu, C., Kershberg, L., Wang, J., Schneeberger, S. & Kaeser, P. S. Dopamine Secretion Is Mediated by Sparse Active Zone-like Release Sites. *Cell* **172**, 706–718 (2018).
21. Liu, C. *et al.* The Active Zone Protein Family ELKS Supports Ca²⁺ Influx at Nerve Terminals of Inhibitory Hippocampal Neurons. *J. Neurosci.* **34**, 12289–12303 (2014).
22. Held, R. G., Liu, C. & Kaeser, P. S. ELKS controls the pool of readily releasable vesicles at excitatory synapses through its N-terminal coiled-coil domains. *Elife* **5**, (2016).
23. Nyitrai, H., Wang, S. S. H. & Kaeser, P. S. ELKS1 Captures Rab6-Marked Vesicular Cargo in Presynaptic Nerve Terminals. *Cell Rep.* **31**, 107712 (2020).
24. Rebola, N. *et al.* Distinct Nanoscale Calcium Channel and Synaptic Vesicle Topographies Contribute to the Diversity of Synaptic Function. *Neuron* **104**, 693-710.e9 (2019).
25. Sakamoto, H. *et al.* Synaptic weight set by Munc13-1 supramolecular assemblies. *Nat. Neurosci.* **21**, 41–49 (2018).
26. Tang, A.-H. *et al.* A trans-synaptic nanocolumn aligns neurotransmitter release to receptors. *Nature* **536**, 210–214 (2016).
27. Sala, K. *et al.* The ERC1 scaffold protein implicated in cell motility drives the assembly of a liquid phase. *Sci. Rep.* **9**, 1–14 (2019).
28. Feric, M. *et al.* Coexisting Liquid Phases Underlie Nucleolar Subcompartments. *Cell* **165**, 1686–1697 (2016).
29. Wei, Z. *et al.* Liprin-mediated large signaling complex organization revealed by the liprin-alpha/CASK and liprin-alpha/liprin-beta complex structures. *Mol Cell* **43**, 586–598 (2011).
30. Ko, J., Na, M., Kim, S., Lee, J. R. & Kim, E. Interaction of the ERC family of RIM-binding proteins with the liprin-alpha family of multidomain proteins. *J. Biol. Chem.* **278**, 42377–42385 (2003).
31. Taru, H. & Jin, Y. The Liprin homology domain is essential for the homomeric interaction of SYD-2/Liprin-alpha protein in presynaptic assembly. *J. Neurosci.* **31**, 16261–16268 (2011).
32. Schoch, S. *et al.* RIM1alpha forms a protein scaffold for regulating neurotransmitter release at the active zone. *Nature* **415**, 321–326 (2002).
33. Spangler, S. A. *et al.* Differential expression of liprin- α family proteins in the brain suggests functional diversification. *J. Comp. Neurol.* **519**, 3040–3060 (2011).
34. Zürner, M. & Schoch, S. The mouse and human Liprin-alpha family of scaffolding proteins: genomic organization, expression profiling and regulation by alternative splicing. *Genomics* **93**, 243–253 (2009).

35. Wang, C.-C., Weyrer, C., Paturu, M., Fioravante, D. & Regehr, W. G. Calcium-Dependent Protein Kinase C Is Not Required for Post-Tetanic Potentiation at the Hippocampal CA3 to CA1 Synapse. *J. Neurosci.* **36**, 6393–402 (2016).
36. Emperador-Melero, J. & Kaeser, P. S. Assembly of the presynaptic active zone. *Curr. Opin. Neurobiol.* **63**, 95–103 (2020).

Reviewers' Comments:

Reviewer #1:

Remarks to the Author:

The authors have made an impressive list of revisions with experiments and text modifications to address the comments raised by all three reviewers. All comments and concerns raised by this reviewer have been adequately addressed. These revisions, in my opinion, have further elevated the quality of the already very strong original manuscript. The authors should be congratulated.

Reviewer #2:

Remarks to the Author:

I found my comments addressed in a satisfactory manner and do support publication of the manuscript.

Reviewer #3:

Remarks to the Author:

The authors have performed an impressive list of new experiments and toned down their conclusion. They have addressed my questions. Congratulations on an excellent paper.

Response for Emperador-Melero et al., “PKC-phosphorylation of Liprin- α 3 triggers phase separation and controls presynaptic active zone structure”, NCOMMS-20-40775-A

Below, we cite the reviewer comments in *black, italic typeface*, add our responses in blue typeface.

Responses to Reviewer #1

The authors have made an impressive list of revisions with experiments and text modifications to address the comments raised by all three reviewers. All comments and concerns raised by this reviewer have been adequately addressed. These revisions, in my opinion, have further elevated the quality of the already very strong original manuscript. The authors should be congratulated.

We thank the reviewer for this positive assessment.

Responses to Reviewer #2

I found my comments addressed in a satisfactory manner and do support publication of the manuscript.

We thank the reviewer for this unconditional recommendation for publication.

Responses to Reviewer #3

The authors have performed an impressive list of new experiments and toned down their conclusion. They have addressed my questions. Congratulations on an excellent paper.

We thank the reviewer for assessing our work positively.